

# Deepening roots can enhance carbonate weathering by amplifying CO₂-enriched recharge

**Hang Wen**[1], **Pamela L. Sullivan**[2], **Gwendolyn L. Macpherson**[3], **Sharon A. Billings**[4], **and Li Li**[1]

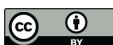CE1 [1]Department of Civil and Environmental Engineering, Pennsylvania State University,
University Park, PA 16802, United States
[2]College of Earth, Ocean, and Atmospheric Science, Oregon State University, Corvallis, OR 97331, United States
[3]Department of Geology, University of Kansas, Lawrence, KS 66045, United States
[4]Department of Ecology and Evolutionary Biology and Kansas Biological Survey, University of Kansas,
Lawrence, KS 66045, United States

**Correspondence:** Li Li (lili@engr.psu.edu)

**Abstract.** TS1 Carbonate weathering is essential in regulating atmospheric CO₂ and carbon cycle at the century timescale. Plant roots accelerate weathering by elevating soil CO₂ via respiration. It however remains poorly understood how and how much rooting characteristics (e.g., depth and density distribution) modify flow paths and weathering. We address this knowledge gap using field data from and reactive transport numerical experiments at the Konza Prairie Biological Station (Konza), Kansas (USA), a site where woody encroachment into grasslands is surmised to deepen roots.

Results indicate that deepening roots can enhance weathering in two ways. First, deepening roots can control thermodynamic limits of carbonate dissolution by regulating how much CO₂ transports vertical downward to the deeper carbonate-rich zone. The base-case data and model from Konza reveal that concentrations of Ca and dissolved inorganic carbon (DIC) are regulated by soil $p$CO₂ driven by the seasonal soil respiration. This relationship can be encapsulated in equations derived in this work describing the dependence of Ca and DIC on temperature and soil CO₂, which applies in multiple carbonate-dominated catchments. Second, numerical experiments show that roots control weathering rates by regulating recharge (or vertical water fluxes) into the deeper carbonate zone and export reaction products at dissolution equilibrium. The numerical experiments explored the potential effects of partitioning 40 % of infiltrated water to depth in woodlands compared to 5 % in grasslands. Soil CO₂ data from woodlands and grasslands suggest relatively similar soil CO₂ distribution over depth and only led to 1 % to 12 % difference in weathering rates if flow partitioning was kept the same between the two land covers. In contrast, deepening roots can enhance weathering by 17 % to 207 % as infiltration rates increased from $3.7 \times 10^{-2}$ to 3.7 m/a. These cases are about 1200 % larger than a case without roots at all, underscoring the essential role of roots in general. Numerical experiments also indicate that weathering fronts in woodlands propagated > 2 times deeper compared to grasslands after 300 years at an infiltration rate of 0.37 m/a. These differences in weathering fronts are ultimately caused by the contact time of CO₂-charged water with abundant carbonate minerals in the deep subsurface. We recognize that modeling results are subject to limitations in representing processes and parameters, but these data and numerical experiments prompt the hypothesis that (1) deepening roots in woodlands can enhance carbonate weathering by promoting recharge and CO₂–carbonate contact in the deep, carbonate-abundant subsurface and (2) the hydrological impacts of rooting characteristics can be more influential than those of soil CO₂ distribution in modulating weathering rates. We call for colocated characterizations of roots, subsurface structure, and soil CO₂ levels, as well as their linkage to water and water chemistry. These measurements will be essential to illuminate feedback mechanisms of land cover changes, chemical weathering, the global carbon cycle, and climate.

# 1 Introduction

Carbonate weathering has long been considered negligible as a long-term control of atmospheric $CO_2$ (0.5 to 1 Ma; Berner and Berner, 2012; Winnick and Maher, 2018). Recent studies, however, have underscored its significance in controlling the global carbon cycle at the century timescale that is relevant to modern climate change, owing to its rapid dissolution, its fast response to perturbations, and the order-of-magnitude-higher carbon store in carbonate reservoirs compared to the atmosphere (Gaillardet et al., 2019). Carbonate weathering is influenced by many factors, including temperature (Romero-Mujalli et al., 2019b), hydrological regimes (Romero-Mujalli et al., 2019a; Wen and Li, 2018), and soil $CO_2$ concentrations (Covington et al., 2015) arising from different vegetation types (Calmels et al., 2014). Rapid alteration to any of these factors, either human or climate induced, may change global carbonate weathering fluxes and lead to a departure from the current global atmospheric $CO_2$ level. This is particularly important given that about 7 %–12 % of the Earth's continental area is carbonate based and about 25 % of the global population completely or partially depend on waters from karst aquifers (Hartmann et al., 2014).

Plant roots have long been recognized as a dominant biotic driver of chemical weathering and the global carbon cycle (Berner, 1992; Beerling et al., 1998; Brantley et al., 2017a). The growth of forests has been documented to elevate soil $pCO_2$ and amplify dissolved inorganic carbon (DIC) fluxes (Berner, 1997; Andrews and Schlesinger, 2001). Rooting structure can influence weathering in two ways. First, rooting systems (e.g., grasslands, shrublands, and woodlands) may affect the distribution of soil carbon (both organic and inorganic), microbe biomass, and soil respiration (Drever, 1994; Jackson et al., 1996; Billings et al., 2018). The relatively deep root distributions of shrublands compared to grasslands may lead to deeper soil carbon profiles (Jackson et al., 1996; Jobbagy and Jackson, 2000), which may help elevate the deep $CO_2$ and acidity that determine carbonate solubility and weathering rates.

Second, plant roots may affect soil structure and hydrological processes. Root trenching and etching can develop porosity (Mottershead et al., 2003; Hasenmueller et al., 2017). Root death and decay can promote the generation of macropores or, more specifically, biopores with connected networks (Angers and Caron, 1998; Zhang et al., 2015). Root channels have been estimated to account for about 70 % of the total described macropores (Noguchi et al., 1997; Beven and Germann, 2013) and for over 70 % of water fluxes through soils (Watson and Luxmoore, 1986). In grasslands, the lateral, dense spread of roots in upper soil layers promotes the formation of horizontally oriented macropores that support near-surface lateral flow (Cheng et al., 2011). Highly dense fine roots also increase the abundance of organic matter and promote granular or sandy texture soil aggregates that facilitate shallow, near-surface water

flow (Oades, 1993; Nippert et al., 2012). In contrast, in shrublands and forests, generally deeper and thicker roots tend to promote a high abundance of macropores and high connectivity to the deep subsurface (Canadell et al., 1996; Nardini et al., 2016), enhancing the drainage of water to the depth (Pawlik et al., 2016).

It is generally known that rooting characteristics vary among plant species and are critical in regulating water budgets, flow paths, and storage (Sadras, 2003; Nepstad et al., 1994; Jackson et al., 1996; Cheng et al., 2011; Brunner et al., 2015; Fan et al., 2017). Existing studies however have primarily focused on the role of soil $CO_2$ and organic acids (Drever, 1994; Lawrence et al., 2014; Gaillardet et al., 2019; Hauser et al., 2020). Systematic studies on coupled effects of hydrological flow paths, flow partitioning, and soil $CO_2$ distribution are missing, owing to the limitation in data that detail rooting effects on flow partitioning and complex hydrological–biogeochemical interactions. Here we ask the following questions: how and to what degree do rooting characteristics influence carbonate weathering when considering both flow partitioning and soil $CO_2$ distribution? Which factor (flow partitioning or soil $CO_2$ distribution) predominantly controls weathering? We hypothesized that deepening roots in woodlands enhance carbonate weathering by promoting deeper recharge and $CO_2$-carbonate contact in the deep, carbonate-abundant subsurface (Fig. 1).

We tested the hypotheses by a series of numerical experiments integrating reactive transport modeling and water chemistry data from an upland watershed in the Konza Prairie Biological Station, a tallgrass prairie and one of the Long-Term Ecological Research (LTER) sites in the US (Fig. S1). We used the calibrated model to carry out numerical experiments for two end-members of vegetation covers, grasslands and woodlands, under flow-partitioning conditions that are characteristic of their rooting structure. These experiments differentiated the impact of biogeochemical and hydrological drivers and bracketed the range of their potential impacts on weathering, thus providing insights on the missing quantitative link between rooting structure, flow paths, and chemical weathering. We recognize that rooting characteristics can have multiple influences on water flow paths and the water budget, for example, via water uptake and transpiration (Sadras, 2003; Fan et al., 2017; Pierret et al., 2016). This study focuses primarily on their potential influence via the alteration of hydrological flow paths.

# 2 Research site and data sources

## 2.1 Site description

Details of the Konza site are in the Supplement and references therein. Here we provide brief information relevant to this work. Konza is a mesic native grassland where experimentally manipulated, long-term burning regimes have

led to woody encroachment in up to 70 % of the catchment area in some catchments. The mean annual temperature and precipitation are 13 °C and ∼ 835 mm, respectively (Tsypin and Macpherson, 2012). The bedrock contains repeating Permian couplets of limestone (1–2 m thick) and mudstone (2–4 m) (Macpherson et al., 2008). The limestone is primarily calcite with traces of dolomite, while the mudstone is dominated by illite, chlorite, and mixed layers of chlorite–illite and chlorite–vermiculite, varying in abundance from major to trace amounts. With an average thickness of 1–2 m in the lowlands, soils mostly have carbonate less than 25 % (Macpherson et al., 2008). Data suggest that the Konza landscape is undergoing a hydrogeochemical transition that coincides with and may be driven by woody encroachment. Parallel to these changes is a detectable decline in streamflow and an increase in weathering rates (Macpherson and Sullivan, 2019), and groundwater $p$CO$_2$ $C$–$Q$ patterns have exhibited chemodynamic patterns (i.e., solute concentrations are sensitive to changes in discharge) for geogenic species (e.g., Mg and Na) in woody-encroached sites compared to grass sites. Sullivan et al. (2019) hypothesized that concentration–discharge relationships may be affected by woody species with deeper roots, which altered flow paths and mineral–water interactions (Fig. 1). We focus on the upland watershed N04d in Konza (Fig. S1) that has experienced a 4-year burning interval since 1990 and has seen considerable woody encroachment and changes in hydrologic fluxes (Sullivan et al., 2019).

## 2.2 Data sources

Daily total meteoric precipitation and evapotranspiration were from the Konza data website (http://lter.konza.ksu.edu/data TS3). Wet chemistry deposition data were from the National Atmospheric Deposition Program (NADP, http://nadp.slh.wisc.edu/ TS4 TS5). Monthly data of soil gases (16, 84, and 152 cm), soil water (17 and 152 cm), and groundwater (366 cm) from 2009 to 2010 were used for this work (Tsypin and Macpherson, 2012) (Fig. 2a). The sampling points were about 30 m away from the stream. More information on field and laboratory methods was included in the Supplement.

## 3 Reactive transport modeling

### 3.1 Base case: 1-D reactive transport model for the Konza grassland (calibrated with field data)

A 1-D reactive transport model was developed using the code CrunchTope (Steefel et al., 2015). The code solves mass conservation equations integrating advective and diffusive/dispersive transport and geochemical reactions. It has been extensively used in understanding mineral dissolution, chemical weathering, and biogeochemical reactions (e.g., Lawrence et al., 2014; Wen et al., 2016; Deng et al., 2017). In this study, the base case had a porosity of 0.48 and a depth

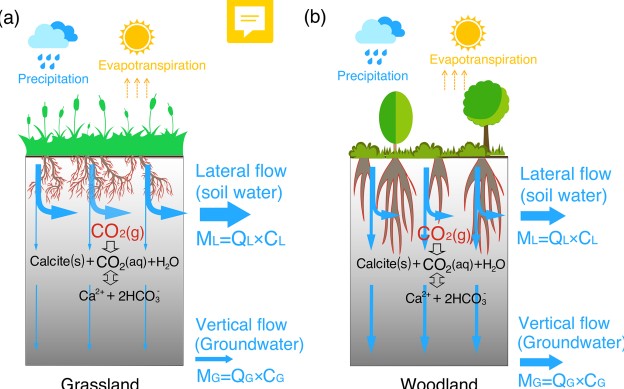

**Figure 1.** A conceptual diagram of hydro-biogeochemical interactions in the grassland **(a)** and woodland **(b)**. The shallow and dense fine roots in the grassland promote lateral macropore development and lateral water flow. In contrast, the woodlands induce vertical macropore development that supports vertical flow (recharge) into the deep, calcite-abundant subsurface compared to the grassland. The gray color gradient reflects the calcite abundance with more calcite in depth. $M_L$ and $M_G$ with a unit of moles per year CE2 (= flow rate $Q$ (m/a) × species concentration $C$ (mol/m$^3$) × unit cross-section area (m$^2$)) represent mass fluxes from lateral flow (soil water) and from vertical flow into groundwater, respectively. Soil water and groundwater were assumed to eventually flow into stream, where infiltration rate $Q_T = Q_L + Q_G$ is the same as discharge that accounts for both lateral and vertical water flow.

of 366.0 cm at a resolution 1.0 cm. Soil temperature was assumed to decrease linearly from 17 °C at the land surface to 8 °C at 366.0 cm, within typical ranges of field measurements (Tsypin and Macpherson, 2012). Detailed setup of domain, soil mineralogy, initial condition, and precipitation chemistry are in the Supplement.

### 3.1.1 Representation of soil CO$_2$

The model does not explicitly simulate soil respiration (microbial activities and root respiration) that produces soil CO$_2$. Instead, it approximates these processes by having a solid phase CO$_2$(g*) that continuously releases CO$_2$(g) that dissolves into CO$_2$(aq) (and other relevant aqueous species) at a kinetic rate constant of $10^{-9}$ mol/m$^2$/s (Reactions 0–1 in Table 1) that reproduced the observed soil $p$CO$_2$ levels. This value is at the low end of the reported soil respiration rates ($10^{-9}$–$10^{-5}$ mol/m$^2$/s) (Bengtson and Bengtsson, 2007; Ahrens et al., 2015; Carey et al., 2016). The dissolution of CO$_2$(g*) via CO$_2$(g) into CO$_2$(aq) follows Henry's law $C_{CO_2}$(aq) $= K_1 p$CO$_2$. Here $K_1$ is the equilibrium constant of Reaction (1), which equals to Henry's law constant. The extent of CO$_2$(g*) dissolution (i.e., the mass change in the solid phase over time) was constrained by $C_{CO_2(aq)}$, which was estimated using temperature-dependent $K_1$ (following the van 't Hoff equation in Eq. S1) and measured soil CO$_2$ data at different horizons (Table 2). These $C_{CO_2(aq)}$

values were then linearly interpolated for individual grid blocks in the model. Finally, these prescribed $C_{CO_2(aq)}$ values were used as the equilibrium constants of the coupled Reactions (0–1) in the form of $CO_2(g^*) \leftrightarrow CO_2(aq)$, such that the soil $CO_2$ values at different depth were represented (Sect. 4.1). In the base case, the soil profile of $C_{CO_2}(aq)$ ~~values~~ was updated monthly based on the monthly soil $CO_2$ data (Table 2). More details about the implementation in Crunch-Tope are included in the Supplement. 

### 3.1.2 Reactions

In the model, the upper soil layers have more anorthite ($CaAl_2Si_2O_8(s)$) and K-feldspar ($KAlSi_3O_8(s)$), and the deeper subsurface contains more calcite (Table S1). The calcite volume increases from 0 % in the upper soil layer to 10 % in the deep subsurface. Table 1 summarizes reactions and thermodynamic and kinetic parameters. Soil $CO_2$ increases pore water acidity (Reactions 0–2) and accelerates mineral dissolution (Reactions 3–6). Silicate dissolution leads to the precipitation of clay (represented by kaolinite in Reaction 7). These reactions were included in the base case to reproduce field data. The kinetics follows the transition state theory (TST) rate law (Plummer et al., 1978) $r = kA\left(1 - \frac{IAP}{K_{eq}}\right)$, where $k$ is the kinetic rate constant (mol/m$^2$/s), $A$ is the mineral surface area per unit volume (m$^2$/m$^3$), IAP is the ion activity product, and $K_{eq}$ is the equilibrium constant. The term IAP / $K_{eq}$ quantifies the extent of disequilibrium: values close to 0 suggest far from equilibrium, whereas values close to 1.0 indicate close to equilibrium.

### 3.1.3 Flow partitioning

Rainwater enter~~ed~~ soil columns at the annual infiltration rate of 0.37 m/a, estimated based on the difference between measured precipitation (0.88 m/a) and evapotranspiration (0.51 m/a). At 50 cm, a lateral flow $Q_L$ (soil water) exited the soil column to the stream at 0.35 m/a. The rest recharged the deeper domain beyond 50 cm (to the groundwater system) at 0.02 m/a ($\sim 2$ % of precipitation) and became groundwater (Fig. 1a), a conservative value compared to 2 %–15 % reported in another study (Steward et al., 2011). The groundwater flow $Q_G$ eventually ~~came out~~ at 366.0 cm and was assumed to enter the stream as part of discharge ($= Q_L + Q_G$). The flow field was implemented in Crunch-Tope using the "PUMP" option.

### 3.1.4 Calibration

We used monthly ~~measured~~ alkalinity and Ca concentration data ~~in different horizons (horizon A and B as well as groundwater in Fig. 2a) from 2009 to 2010 at the Konza grassland~~ for model calibration. The monthly Nash–Sutcliffe efficiency (NSE)~~, which~~ quantified the residual variance of modeling output compared to measurements, was used for model performance evaluation (Moriasi et al., 2007). NSE values higher than 0.5 are considered acceptable. ~~At Konza, calcite in the upper soil (above horizon B) is mixed with other minerals, has small particle size, and is considered impure (Macpherson and Sullivan, 2019) and therefore has a lower $K_{eq}$ value than those at depth with relatively pure calcite. The $K_{eq}$ of impure calcite ($K_4$) was calibrated by fitting field data of Ca and alkalinity.~~

### 3.2 Numerical experiments

Numerical experiments were set up ~~to understand how and to what extent roots regulate weathering rates and solute transport in~~ grasslands and ~~in~~ woodlands. The base case from Konza was used to represent grasslands~~,~~ and the Calhoun site (the Calhoun Critical Zone Observatory in South Carolina, USA) was used as ~~a~~ representative woody site (Billings et al., 2018). Grasslands are typically characterized by a high proportion of horizontal macropores induced by dense, lateral-spread of roots mostly at depths less than 0.8 m (Jackson et al., 1996; Frank et al., 2010). These characteristics promote lateral flow ($Q_L$) at the shallow subsurface (Fig. 1a). At Konza, over 90 % of grass roots were at the top 0.5 m, leading to high hydrologic conductivity in top soils (Nippert et al., 2012). In the woodland, a greater proportion of deep roots enhances vertical macropore development (Canadell et al., 1996; Nardini et al., 2016), reduces permeability contrasts at different depths (Vergani and Graf, 2016), and is thought to facilitate more vertical water flow to the depth ($Q_G$). At the Calhoun site, over 50 % of roots are in the top 0.5 m in woodlands, with the rest penetrating deeper (Jackson et al., 1996; Eberbach, 2003; Billings et al., 2018).

The experiments aimed to compare the general, averaged behaviors rather than event-scale dynamics so the annual-average soil $CO_2$ data and corresponding prescribed $CO_2(aq)$ concentrations were used (Table 2). The experiments focused on calcite weathering (Reactions 0–4) and excluded silicate weathering reactions (Reactions 5–7). The mineral-dissolution ~~related~~ parameters from the base case were used for all experiments. We compared the relative significance of the hydrological (i.e., lateral versus vertical flow partitioning) vs. respiratory (i.e., $CO_2$ generation) influences of deepening roots. Though other potential differences might be induced by deepening roots (e.g., water uptake, water table, and transpiration), we assumed they remain constant across the grassland and woodland simulations to examine the relative influences of flow path vs. $CO_2$ generation on carbonate weathering.

### 3.2.1 Hydrological ~~and~~ biogeochemical differences in grasslands and woodlands

Flow partitioning between lateral shallow flow and vertical recharge flow is challenging to quantify and is subject to large uncertainties under diverse climate, lithology, and land

**Table 1.** Key reactions and kinetic and thermodynamic parameters.

| Reaction | $\log_{10} K_{eq}$ at 25 °C[a] | Standard enthalpy ($\Delta H^\circ$, kJ/mol) | $\log_{10} k$ (mol/m²/s) at 25 °C[d] | Specific surface area (m²/g)[d] |
|---|---|---|---|---|
| **Soil CO₂ production through CO₂ (g*) and dissolving into CO₂(aq)** | | | | |
| (0) $CO_2(g^*) \leftrightarrow CO_2(g)$ | – | – | −9.00 | 1.0 |
| (1) $CO_2(g) \leftrightarrow CO_2(aq)$ | −1.46[b] | −19.98 | – | – |
| (2) $CO_2(aq) + H_2O \leftrightarrow H^+ + HCO_3^-$ | −6.35 | 9.10 | – | – |
| (3) $HCO_3^- \leftrightarrow H^+ + CO_3^{2-}$ | −10.33 | 14.90 | – | – |
| **Chemical weathering** | | | | |
| (4) $CaCO_3(s) + CO_2(aq) + H_2O \leftrightarrow Ca^{2+} + 2HCO_3^-$ | −5.12[c] −4.52[c] | −15.41 | −6.69 | 0.84 |
| (5)[e] $CaAl_2Si_2O_8(s) + 8H^+ \leftrightarrow Ca^{2+} + 2Al^{3+} + 2H_4SiO_4(aq)$ | 26.58 | – | −11.00 | 0.045 |
| (6)[e] $KAlSi_3O_8(s) + 4H^+ + 4H_2O \leftrightarrow Al^{3+} + K^+ + 3H_4SiO_4(aq)$ | −0.28 | – | −12.41 | 0.20 |
| (7)[e] $Al_2Si_2O_5(OH)_4(s) + 6H^+ \leftrightarrow 2Al^{3+} + 2H_4SiO_4(aq) + H_2O$ | 6.81 | – | 12.97 | 17.50 |

[a] Values of $K_{eq}$ were interpolated using the EQ3/6 database (Wolery et al., 1990), except Reactions (1) and (4) (i.e., $K_1$ and $K_4$). [b] By combining the temporally and spatially dependent soil $pCO_2$ data, $CO_2(aq)$ concentrations were calculated through $C_{CO_2}(aq) = K_1 pCO_2$. The prescribed $C_{CO_2}(aq)$ values were used as equilibrium constants in CrunchTope to describe how much soil CO₂ was available for weathering. [c] Calcite in the upper soil (above horizon B) is mixed with other minerals and has small particle size (Macpherson and Sullivan, 2019), is relatively impure, and therefore has a lower $K_{eq}$ value than those at depth with relatively pure calcite. The $K_{eq}$ of the impure calcite was calibrated by fitting field data of Ca and alkalinity. [d] The kinetic rate parameters and specific surface areas were from Palandri and Kharaka (2004), except Reaction (0). The kinetic rate constant of the solid phase CO₂(g*) dissolution (i.e., soil CO₂ production rate constant) was from Bengtson and Bengtsson (2007), Ahrens et al. (2015), and Carey et al. (2016); the specific surface area was referred to that of soil organic carbon (Pennell et al., 1995). [e] These reactions were only used in the base-case model as they occurred in upper soils in Konza. In the later numerical experiments, these reactions were not included so as to focus on carbonate weathering.

cover conditions. The ratios of lateral flow in upper soils versus the total flow inferred from a tracer study in a grassland vary from ∼ 70 % to ∼ 95 % (Weiler and Naef, 2003). Harman and Cosans (2019) found that the lateral flow rate at upper soils over the overall infiltration can vary between 50 % and 95 %. Deeper roots in woodlands can increase deep soil permeability by over 1 order of magnitude (Vergani and Graf, 2016). Assuming that the vertical, recharged flow water ultimately leaves the watershed as baseflow, the ratio of the lateral versus vertical flow has been reported with a wide range. In forests such as Shale Hills in Pennsylvania and Coal Creek in Colorado, ∼ 7 %–20 % of stream discharge is from groundwater, presumably recharged by vertical flow (Li et al., 2017; Zhi et al., 2019). Values of $Q_G / Q_T$ estimated through base flow separation vary from 20 % to 90 % in forest/wood-dominated watersheds (Price, 2011), often negatively correlating with the proportion of grasslands (Mazvimavi et al., 2004).

Soil respiration rates can vary between $10^{-9}$ and $10^{-5}$ mol/m²/s for both grasslands and woodlands (Bengtson and Bengtsson, 2007; Ahrens et al., 2015; Carey et al., 2016). Soil CO₂ levels may vary by 2–3 orders of magnitude depending on vegetation type and climate conditions (Neff and Hooper, 2002; Breecker et al., 2010). Soil CO₂ levels are further complicated also by their dependence on topographic position, soil depth, and soil moisture, all of which

determine the magnitude of microbial and root activities and CO₂ diffusion (Hasenmueller et al., 2015; Billings et al., 2018). There is however no consistent evidence suggesting which land cover exhibits higher soil respiration rate or soil CO₂ level. Below we describe details of the numerical experiments (Table 3) exploring the influence of hydrological versus biogeochemical impacts of roots.

### 3.3 Three scenarios

Each scenario in Table 3 includes a grassland and woodland case, with their respective profiles of calcite distribution and soil $pCO_2$ kept the same (column 3 and 4 in Table 3). The only difference in different scenarios is the flow partitioning (column 5 in Table 3). Soil $pCO_2$ in the grassland (red line in Table 3) was set to reflect the annual average of the Konza site. Soil $pCO_2$ (blue line) in the woodland is the annual average from the forest-dominant Calhoun site (Billings et al., 2018). In all scenarios, we assumed that grasslands and woodlands had the same total solid phase CO₂(g*) producing CO₂ gas and CO₂(aq) but differed in depth distributions. The CO₂(g*) served as the source of soil CO₂ and was constrained by CO₂(g) field data (Table 2). In grasslands, the modeled distribution of CO₂(g*) was steep, with higher density of roots and more abundant CO₂(g*) in the upper soils, while the distribution was less steep in woodlands (Jackson

**Table 2.** Measured $CO_2(g)$ at different depths and corresponding estimated $C_{CO_2}(aq)$.

| Time | Soil depth | | | | Ways obtained[a,b,c] |
|---|---|---|---|---|---|
| **I. Grassland (Konza) cases** | | | | | |
| | Horizon A ($h = 16$ cm) | Horizon AB (84 cm) | Horizon B (152 cm) | Groundwater (366 cm) | |
| Soil $T$ °C | 17 | 15 | 13 | 8 | Estimated |
| $K_1$ | $4.1 \times 10^{-2}$ | $4.4 \times 10^{-2}$ | $4.6 \times 10^{-2}$ | $5.4 \times 10^{-2}$ | Estimated |
| **1. Base case with monthly ~~updated~~ $CO_2(g)$ (%) and $CO_2(aq)$ (mol/L)** | | | | | |
| $CO_2(g)$   July | 3.6 | 6.8 | 6.6 | 2.2 | Measured |
| August | 1.4 | 1.7 | 7.2 | 3.9 | Measured |
| September | 0.6 | 1.2 | 3.9 | 4.9 | Measured |
| October | 0.5 | 1.4 | 2.5 | 5.0 | Measured |
| November | 0.6 | 1.1 | 2.2 | 4.0 | Measured |
| January | 0.3 | 0.8 | 1.1 | 3.6 | Measured |
| March | 0.2 | 0.3 | 0.5 | 3.0 | Measured |
| $CO_2(aq)$   July | $1.5 \times 10^{-3}$ | $3.0 \times 10^{-3}$ | $3.0 \times 10^{-3}$ | $1.2 \times 10^{-3}$ | Estimated |
| August | $5.7 \times 10^{-4}$ | $7.5 \times 10^{-4}$ | $3.3 \times 10^{-3}$ | $2.1 \times 10^{-3}$ | Estimated |
| September | $2.5 \times 10^{-4}$ | $6.2 \times 10^{-4}$ | $1.8 \times 10^{-3}$ | $2.6 \times 10^{-3}$ | Estimated |
| October | $1.9 \times 10^{-4}$ | $6.2 \times 10^{-4}$ | $1.2 \times 10^{-3}$ | $2.7 \times 10^{-3}$ | Estimated |
| November | $2.5 \times 10^{-4}$ | $4.8 \times 10^{-4}$ | $1.0 \times 10^{-3}$ | $2.2 \times 10^{-3}$ | Estimated |
| January | $1.2 \times 10^{-4}$ | $3.5 \times 10^{-4}$ | $5.1 \times 10^{-4}$ | $1.9 \times 10^{-3}$ | Estimated |
| March | $8.2 \times 10^{-5}$ | $1.3 \times 10^{-4}$ | $2.3 \times 10^{-4}$ | $1.6 \times 10^{-3}$ | Estimated |
| **2. Numerical experiments with annual-average $CO_2(g)$ (%) and $CO_2(aq)$ (mol/L)** | | | | | |
| $CO_2(g)$   Annual | $1.0 \pm 1.2$ | $1.9 \pm 2.2$ | $3.4 \pm 2.6$ | $3.8 \pm 1.0$ | Measured |
| $CO_2(aq)$   Annual | $(4.2 \pm 5.0) \times 10^{-4}$ | $(8.5 \pm 9.7) \times 10^{-4}$ | $(1.6 \pm 1.2) \times 10^{-3}$ | $(2.0 \pm 0.5) \times 10^{-3}$ | Estimated |
| **II. Woodland cases (Calhoun site, South Carolina) numerical experiments with annual-average $CO_2(g)$ (%) and $CO_2(aq)$ (mol/L)** | | | | | |
| | ($h = 50$ cm) | (150 cm) | (300 cm) | (500 cm) | |
| Soil $T$ °C | 16 | 13 | 10 | 8 | Estimated |
| $K_1$ | $4.2 \times 10^{-2}$ | $4.6 \times 10^{-2}$ | $4.9 \times 10^{-2}$ | $5.4 \times 10^{-2}$ | Estimated |
| $CO_2(g)$   Annual | $0.9 \pm 0.1$ | $2.7 \pm 0.6$ | $3.8 \pm 0.6$ | $3.9 \pm 0.7$ | Measured |
| $CO_2(aq)$   Annual | $(4.5 \pm 0.7) \times 10^{-4}$ | $(1.3 \pm 0.3) \times 10^{-3}$ | $(1.9 \pm 0.3) \times 10^{-3}$ | $(1.9 \pm 0.3) \times 10^{-3}$ | Estimated |

[a] Monthly measured soil $CO_2$ data for the Konza grassland (base case) were from Tsypin and Macpherson (2012); the ~~annual-average~~ soil $CO_2$ ~~used~~ in the grassland experiments was averaged from ~~the corresponding~~ monthly measurements. More information on measurements at the Konza grassland ~~was detailed~~ in the Supplement. The annual-average soil $CO_2$ in the woodland experiments was from the Calhoun site in South Carolina ~~where trees (forest) are dominant~~ (Billings et al., 2018). [b] $CO_2(aq)$ values were estimated using Henry's law: $C_{CO_2}(aq) = K_1 pCO_2$; the temperature-dependent $K_1$ was calculated following Eq. (S1). The $C_{CO_2}(aq)$ values at different soil depths were used to prescribe the available soil $CO_2$ for chemical weathering. [c] Soil temperature was estimated from the soil water and shallow groundwater temperature (Tsypin and Macpherson, 2012; Billings et al., 2018).

et al., 1996). Below the rooting depth, $CO_2(g*)$ was assumed to be smaller (by 10 times) to represent the potential soil $CO_2$ sources from microbial activities (Billings et al., 2018).

Scenario 1 considered flow partitioning (Table 3). With the large permeability contrast of soil and bedrock (over 4 orders of magnitude) in Konza (Macpherson, 1996), the $Grass_{PF}$ case (with $Q_L = 95\% \times Q_T$ and $Q_G = 5\% \times Q_T$) represents an end-member case for the grassland. The woodland ($Wood_{PF}$) case was set to have 60 % lateral flow and 40 % vertical flow into the deeper subsurface. This groundwater percentage is at the high end of flow partitioning and serves as an end-member case for woodlands (Vergani and Graf, 2016). Scenarios 2 and 3 had no flow partitioning. Scenario 2 had two cases with 100 % vertical flow via the bottom outlet (VF; $Wood_{VF}$ and $Grass_{VF}$); Scenario 3 had two cases with 100 % horizontal flow (HF; $Wood_{HF}$ and $Grass_{HF}$) via the shallow outlet at 50 cm. These cases represent the end-member flow cases with 100 % lateral flow or 100 % vertical flow. In addition, because the two cases have the same flow scheme, they enable the differentiation of effects of soil $CO_2$

*Please note the remarks at the end of the manuscript.*

**Table 3.** Physical and geochemical characteristics in numerical CE3

| | Cases | Calcite distribution[b] | Schematic | Flow partitioning[c] | Soil $CO_2$ distribution[d] |
|---|---|---|---|---|---|
| | | | | **Hydrological and biogeochemical aspects** | |
| Scenario 1 with flow partitioning (PF) | $Grass_{PF}$ | | | $Q_L = 95\% \, Q_T$<br>$Q_G = 5\% \, Q_T$ (red arrow) | Annual average from the Konza grassland (red line) |
| | $Wood_{PF}$ | | | $Q_L = 60\% \, Q_T$<br>$Q_G = 40\% \, Q_T$ (blue arrow) | Annual average from the Calhoun forest (blue line) |
| Scenario 2 with 100% vertical flow (VF) | $Grass_{VF}$ | | | $Q_L = 0$<br>$Q_G = Q_T$ (black arrow) | Annual average from the Konza grassland (red line) |
| | $Wood_{VF}$ | | | | Annual average from the Calhoun forest (blue line) |
| Scenario 3 with 100% lateral flow (HF) | $Grass_{HF}$ | | | $Q_L = Q_T$<br>$Q_G = 0$ (black arrow) | Annual average from the Konza grassland (red line) |
| | $Wood_{HF}$ | | | | Annual average from the Calhoun forest (blue line) |

a. Diagrams represent the soil profiles of physical and geochemical properties.
b. Calcite distribution (black line) in all cases increases with depth; the values are listed in Table S1 and shown in Figure 3A.
c. The flow field was implemented in CrunchTope using the "PUMP" option.
d. Annual-average soil $CO_2$ used for constraining the grassland (red line) was from the Konza grassland (Tsypin and Macpherson, 2012). The annual-average soil $CO_2$ in the woodland (blue line) was from the Calhoun forest (Billings et al., 2018).

distribution versus hydrology differences. All scenarios were run under infiltration rates from $3.7 \times 10^{-2}$ to 3.7 m/a ($10^{-4}$–$10^{-2}$ m/d), the observed daily variation range at Konza. This was to explore the role of flow regimes and identify conditions where the most and least significant differences occur. ~~To reproduce the observed soil $CO_2$ profile, the soil $CO_2$ production rate (mol/m²/a) at the domain scale (calculated by the mass change in the solid phase $CO_2(g*)$ over time) was assumed to increase with infiltration rates (Fig. S2). This is consistent with field observations that soil $CO_2$ production rate and efflux may increase with rainfall in grassland and forest ecosystems (Harper et al., 2005; Patrick et al., 2007; Wu et al., 2011; Vargas et al., 2012; Jiang et al., 2013). For example, Zhou et al. (2009) documented soil $CO_2$ production rates increasing from 3.2 to 63.0 mol/m²/a when the annual precipitation increased from 400 to 1200 mm. In addition, Wu et al. (2011) showed that increasing precipitation from 5 to 2148 mm enhanced soil respiration by 40% and that a global increase of 2 mm precipitation per decade may lead to an increase of 3.8 mol/m²/a for soil $CO_2$ production. The simulated soil $CO_2$ production rates across different infiltration rates here (~0.1–10 mol/m²/a shown in Fig. S2) were close to the reported belowground net primary production (belowground NPP) of typical ecosystems: ~0.8–100 mol/m²/a for grasslands (Gill et al., 2002) and ~0.4–40 mol/m²/a for woodlands (Aragao et al., 2009 TS4).~~

Each case was run until steady state, when concentrations at the domain outlet became constant (within ±5%) over time. The time to reach steady state varied from 0.1 to 30 a, depending on infiltration rates. The lateral flux (soil water, $M_L = Q_L \times C_L$) and vertical fluxes (groundwater, $M_G = Q_G \times C_G$) were calculated at 50 and 366 cm, in addition to total fluxes (weathering rates, $M_T$). These fluxes multiplied

with unit cross-section area (m²) convert into rates in units of moles per year (mol/a).

### 3.4 Carbonate weathering over century timescales: soil property evolution

To compare the propagation of weathering fronts over longer timescales, we carried out two 300-year simulations for Scenario 1 with flow partitioning under the base-case infiltration rate of 0.37 m/a (i.e., $Grass_{PF}$ and $Wood_{PF}$ in Table 3). During this long-term simulation, we updated calcite volume, porosity, and permeability. The calcite volume changes were updated in each time step based on corresponding mass changes, which were used to update porosity. Permeability changes were updated based on changes in local porosity following the Kozeny–Carman equation: $\frac{k_i}{k_{i,0}} = \left(\frac{\varnothing_i}{\varnothing_{i,0}}\right)^3 \times \left(\frac{1-\varnothing_{i,0}}{1-\varnothing_i}\right)^2$ (Kozeny, 1927; Costa, 2006), where $k_i$ and $\varnothing_i$ are permeability and porosity in grid $i$ at time $t$, and $k_{i,0}$ and $\varnothing_{i,0}$ are the initial permeability and porosity, respectively.

### 3.5 Quantification of weathering rates and their dependence on $CO_2$–carbonate contact

To quantify the overall weathering rates and $CO_2$–carbonate contact in each scenario, we used the framework from a previously developed upscaled rate law for dissolution of spatially heterogeneously distributed minerals (Wen and Li, 2018). The rate law says that three characteristic times are important. The equilibrium time $\tau_{eq}$ represents the characteristic timescale of mineral dissolution to reach equilibrium in a well-mixed system. The residence time $\tau_a$, i.e., the timescale of advection, quantifies the overall water contact

time with the whole domain. It was calculated by the product of domain length ($L$) and porosity divided by the overall infiltration rate ($Q_T$): $\tau_a = \frac{L\phi}{Q_T}$. The reactive transport time $\tau_{ad,r}$ quantified the water contact time with calcite as influenced by both advection and diffusion/dispersion. The upscaled rate law is as follows:

$$R_{calcite} = k_{calcite} A_T \left[ 1 - \exp\left( -\frac{\tau_{eq}}{\tau_a} \right) \right]$$
$$\left\{ 1 - \exp\left[ -L\left( \frac{\tau_a}{\tau_{ad,r}} \right) \right] \right\}^\alpha \qquad\qquad . \qquad (1)$$

Here $k$ is the intrinsic rate constant measured for a mineral in a well-mixed reactor, $A_T$ is the total surface area, $L$ is domain length, $\alpha$ is geostatistical characteristics of spatial heterogeneity, and $\frac{\tau_a}{\tau_{ad,r}}$ is the reactive time ratio quantifying the relative magnitude of the water contact time with the whole domain versus the contact time with the reacting mineral. This rate law consists of two parts: the effective dissolution rates in homogeneous media represented by $kA_T \left[ 1 - \exp\left( -\frac{\tau_{eq}}{\tau_a} \right) \right]$ and the heterogeneity factor $\left\{ 1 - \exp\left[ -L\left( \frac{\tau_a}{\tau_{ad,r}} \right) \right] \right\}^\alpha$ that quantifies effects of preferential flow paths arising from heterogeneous distribution of minerals. When $\frac{\tau_a}{\tau_{ad,r}} > 1$, the water contact time with calcite zone is small, meaning the water is replenished quickly compared to the whole domain, leading to higher $CO_2$–carbonate interactions. In contrast, a small $\frac{\tau_a}{\tau_{ad,r}}$ ratio ($< 1$) reflects that water is replenished slowly in the reactive calcite zones, leading to less $CO_2$–carbonate contact. These different timescales were calculated for Scenario 1–3 based on the flow characteristics and dissolution thermodynamics and kinetics, as detailed in the Supplement. Values of $\tau_{eq}$, $\tau_a$, and $\tau_{ad,r}$ for all experiments are listed in Table S2. Note that numerical experiments in Scenario 1–3 focused on the short-term scale, with negligible changes in the solid phase.

## 4   Results

### 4.1   The thermodynamics of carbonate dissolution: grassland at Konza as the base-case scenario

The calibrated model reproduced the temporal dynamics with a Nash–Sutcliffe efficiency (NSE) value $> 0.6$ and was considered satisfactory (Fig. 2). Note that the $y$ axis is inverted to display upper soils at the top and deep soils at the bottom to be consistent with their subsurface position shown in Fig. 2a. The measured $CO_2(g)$ varied between 0.24 % and 7.30 % (Fig. 2b right axis), 1 to 2 orders of magnitude higher than the atmospheric level of 0.04 %. The estimated $CO_2(aq)$ (Fig. 2b left axis) generally increased with depth except in July and August when horizon B was at peak concentration. The timing of the peaks and valleys varied in different horizons. The $CO_2(aq)$ reached maxima in summer in soil horizons A and B and decreased to less than 0.5 mM in winter

and spring. The groundwater $CO_2(aq)$ exhibited a delayed peak in September and October and dampened seasonal variation compared to soil horizons. The temporal trends of alkalinity, DIC, and Ca in groundwater mirrored those of soil $CO_2$ at their corresponding depths, indicating the predominate control of soil $CO_2$ on carbonate weathering. The groundwater concentrations of these species were also higher than soil concentrations. The simulated groundwater DIC (approximately summation of $CO_2(aq)$ and alkalinity) was $> 6$ times higher than that in upper soil ($\sim 1.0$ mM). The dissolved mineral volume was negligible for the simulation period ($< 0.5 \% v/v$). Sensitivity analysis revealed that changes in flow velocities influenced concentrations in horizon A ~~with~~ anorthite ~~as~~ the dominating dissolving mineral (0–1.8 m); their effects are negligible in horizon B and groundwater where fast-dissolving calcite ~~has reached~~ equilibrium.

Several measurements/parameters were important in reproducing data~~, especially~~ soil $CO_2$, which determined the $CO_2(aq)$ level and its spatial variation, and equilibrium constant ($K_{eq}$) of calcite dissolution. Imposition of monthly variations and depth distributions of soil $CO_2$ were essential to capture the variation of alkalinity and Ca data at different horizons. The imposition of calcite $K_{eq}$ was also critical for reproducing Ca concentrations. Impurities were suggested to affect $K_{eq}$ of natural calcite by a factor of $\sim 2.0$ (Macpherson and Sullivan, 2019). Calcite $K_{eq}$ in upper soils had to be reduced by a factor of 3.8 in the model to reproduce concentrations in horizon A. The alkalinity and Ca concentrations were not sensitive to kinetic parameters nor precipitation, because carbonate dissolution ~~was fast and thermodynamically controlled.~~

### 4.2   Numerical experiments: the significance of ~~potential hydrology differences~~

*Scenario 1 for hydro-biogeochemical effects with flow partitioning (Wood$_{PF}$ and Grass$_{PF}$).* Figures 3A1–E1 show depth profiles of calcite and soil $CO_2$ production rates and steady-state concentrations of reaction products. ~~Although the model did not explicitly simulate soil respiration, the~~ soil $CO_2$ production rate was highest in the upper soil at around $10^{-6.5}$ mol/s and decreased to $\sim 10^{-9.5}$ mol/s at 366 cm (Fig. 3B1), consistent with the decline with soil depth observed in natural systems. The $CO_2(g)$ level (released from ~~the solid phase~~ $CO_2(g^*)$) increased with depth due to ~~autotrophic and heterotrophic respiration~~ and downward fluxes of $CO_2$–charge water from the upper soil (Fig. 3C1). Concentrations of reaction products (Ca, DIC) were lower in the top 40 cm, reflecting the lower carbonate-mineral background level, and higher at depths over 60 cm. The transition occurred between 35 and 60 cm in the vicinity of the calcite–no-calcite interface at 55 cm, where concentrations of Ca and DIC increased abruptly until reaching equilibrium. This thin transition was driven by fast calcite dissolution and

**Biogeosciences, 17, 1–20, 2020**                                                **https://doi.org/10.5194/bg-17-1-2020**

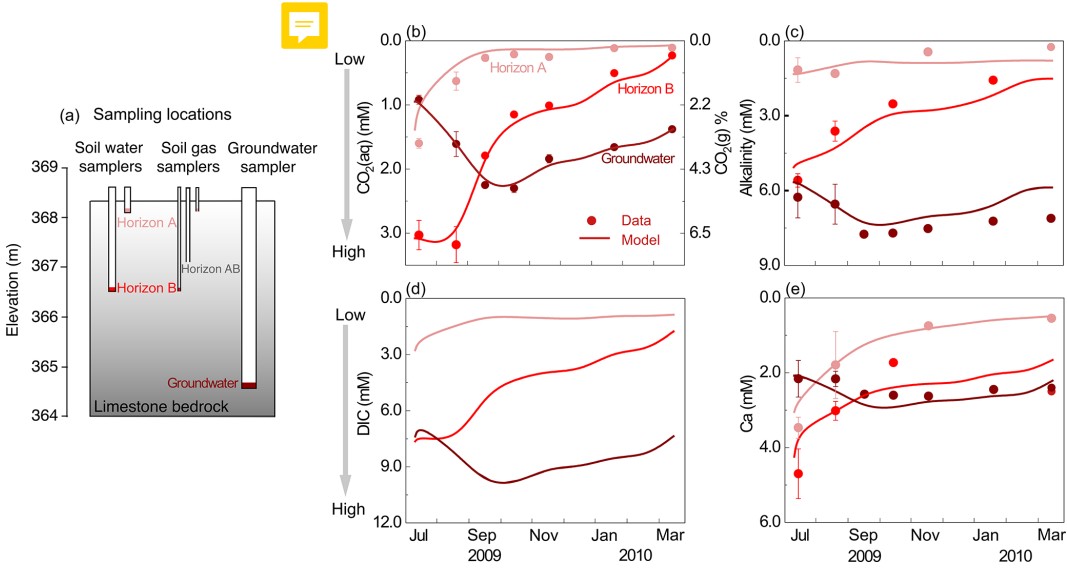

**Figure 2. (a)** Schematic representation of sampling depths in the Konza grassland; corresponding monthly dynamics of **(b)** $CO_2(aq)$ and $CO_2(g)$, **(c)** alkalinity, **(d)** DIC, and **(e)** Ca concentrations. Lines represent modeling outputs at the corresponding sampling depth of monthly field measurements (dots), including soil water at horizons A (16 cm) and B (152 cm), as well as groundwater (366 cm). The lines of $CO_2(aq)$ and $CO_2(g)$ in panel **(b)** overlapped. Note that there are no DIC data so no dots in panel **(d)**. The temporal trends of alkalinity, DIC, and Ca mirrored those of soil $CO_2$, indicating its predominant control on weathering. ~~Note that concentrations on $y$ axis from **(b–e)** are low to high, to be consistent with the layout in panel **(a)** with deeper water chemistry showing up deeper.~~ TS7

rapid approach to equilibrium, resulting in a short equilibrium distance. The equilibrated DIC and Ca concentrations below 60 cm followed the similar increasing trend of $CO_2(g)$ with depth in the deep zone (Fig. 3C1–E1).

Figure 3F1 shows that Ca concentrations in soil water (light color) were lower than groundwater (dark color) and varied with infiltration rates. The difference between soil water and groundwater Ca concentrations was relatively small at low infiltration rates because both reached equilibrium but diverged at high infiltration rates. Higher infiltration rates diluted soil water but not as much for groundwater. As expected, stream ~~concentrations~~, a mixture of soil water and groundwater (solid line), were in between these values but closely resembled soil water in the grassland. In both cases, stream concentration decreased as infiltration increased, indicating a dilution concentration–discharge relationship.

*Scenario 2–3 for biogeochemical effects (without flow partitioning).* Scenarios 2 and 3 were end-member cases that bracketed the range of rooting effects. Calcite and soil $CO_2$ were distributed the same way as their corresponding flow partitioning (CL PF) cases (Table 3). The $Wood_{VF}$ and $Grass_{VF}$ cases had 100 % flow going downward via the deeper calcite zone maximizing the $CO_2$–water–calcite contact. In the $Wood_{HF}$ and $Grass_{HF}$ cases (Table 3), all water exited at 50 cm, bypassing the deeper calcite-abundant zone and minimizing the $CO_2$–water–calcite contact. Figure 3A2–F2 and 3A3–F3 show the VF ($Wood_{VF}$ and $Grass_{VF}$) and HF ($Wood_{HF}$ and $Grass_{HF}$) cases, respectively. Similar to the flow partitioning cases, the concentrations of reaction prod-

ucts were low in the shallow zone and increased over 10 times within a short distance $\sim 5$ cm at the depth of $\sim 50$ cm. The woodland cases increased slightly more than the grassland cases because of the slightly steeper soil $CO_2$ distribution (Fig. 3C2 and 3C3). Figure 3F2 indicates that the effluent Ca concentrations were slightly higher in $Wood_{VF}$ due to the high soil $CO_2$ level at the bottom outlet. The $Grass_{HF}$ and $Wood_{HF}$ case almost had the same effluent Ca concentrations at the upper soil (Fig. 3F3).

*Concentration–discharge relationship and weathering rates in all cases.* The VF cases had the highest effluent concentrations and weathering rates, whereas the HF cases had lowest concentrations and weathering rates, and the $Grass_{PF}$ and $Wood_{PF}$ cases fell in between (Fig. 4a, b). This is because the VF cases maximized the $CO_2$–calcite contact with 100 % flow through the calcite zone, whereas the HF cases had minimum $CO_2$–calcite contact with water bypassing the calcite zone (column 3–4 in Table 3). The $Grass_{PF}$ and $Wood_{PF}$ cases ~~allowed~~ different extent of contact prescribed by the amount of flow via the calcite zone. A case run without soil respiration (i.e., no $CO_2(g^*)$, not shown) indicated that Ca and DIC concentrations were more than an order of magnitude lower than cases with soil respiration. In addition, the PF and HF cases generally showed dilution patterns with concentration decreasing with infiltration rates, as compared to the VF cases where a chemostatic pattern emerges with almost no changes ~~with~~ infiltration rates. This is because in the VF cases the concentrations mostly reached equilibrium concentration. In the PF and HF cases, a large proportion of

Please note the remarks at the end of the manuscript.

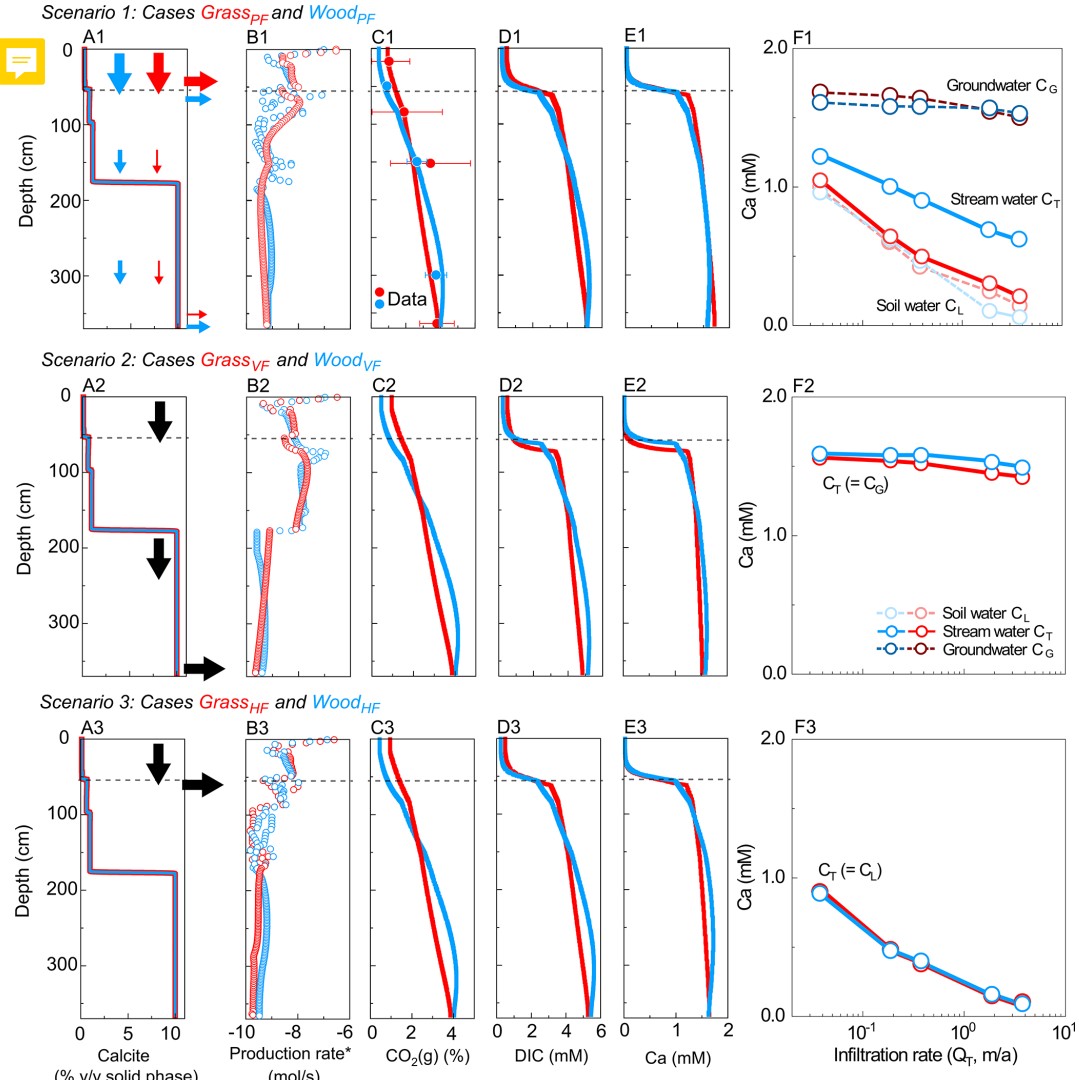

**Figure 3.** Simulated depth profiles of (A) calcite (vol %) and (B) soil $CO_2$ production rate; (C) $CO_2(g)$ (%), (D) DIC, and (E) Ca at 0.37 m/a; and (F) effluent Ca concentrations at different infiltration rates. From top to bottom rows are Scenario 1 ($Grass_{PF}$ and $Wood_{PF}$, with flow partitioning, first row), Scenario 2 ($Grass_{VF}$ and $Wood_{VF}$, with 100 % vertical flow, second row), and Scenario 3 ($Grass_{HF}$ and $Wood_{HF}$, with 100 % lateral flow, last row). Red and blue colors present grassland and woodland, respectively. Lines and empty circles represent modeling outputs, while filled circles with error bar in C are the annual-average $CO_2(g)$ data (in Table 2). Arrows in A indicate flow conditions. In F, soil water and groundwater refer to concentrations at the lateral (50 cm) and vertical (366 cm) outlets, respectively. Higher fractions of vertical flow in $Wood_{PF}$ led to higher stream Ca concentration compared to $Grass_{PF}$. In Scenario 2 and 3 ~~without flow partitioning~~, stream Ca concentrations were similar. The concentrations of DIC versus discharge are very similar to the Ca concentrations in F.

water flows through soils with negligible calcite where the water moves away from equilibrium as infiltration rates increase.

Although weathering rates generally increased with infiltration rates, the woodland increased more ($4.6 \times 10^{-2}$ to 2.4 mol/a in $Wood_{PF}$), ~~Estimates of the overall soil $CO_2$ production rate in $Grass_{PF}$ and $Wood_{PF}$ suggest that the values under the same infiltration rate were all within a difference of 10 % (Fig. S2), indicating that the differences in weathering rates are mainly driven by the flow partitioning.~~ As a refer-

~~ence, the weathering rate in cases with soil respiration was up to 10 times higher compared to that without soil respiration.~~

*Development of reaction fronts at the century scale.* To explore the longer-term effects, we ran the PF cases at 0.37 m/a for 300 years. More water flowing vertically through abundant calcite zones in $Wood_{PF}$ resulted in faster weathering and a deeper reaction front at a depth of $\sim 210$ cm compared to $\sim 95$ cm in $Grass_{PF}$ (Fig. 5a). The depletion of calcite led to an increase in porosity (Fig. 5b), which was over 1 order of magnitude higher at the domain scale of the woodland

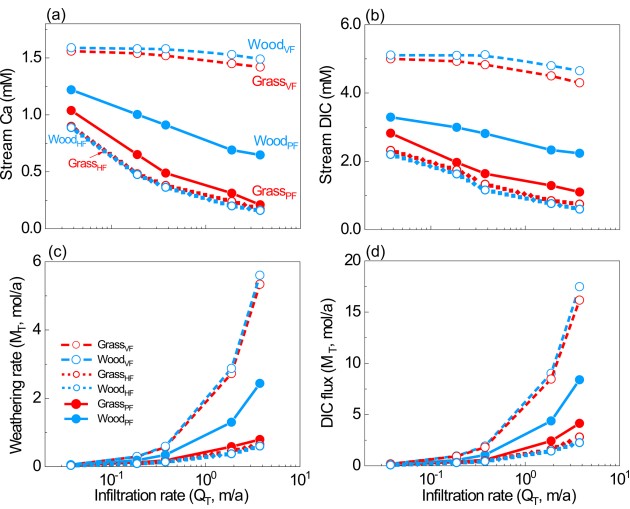

**Figure 4. (a–b)** Stream ~~water~~ Ca and DIC concentrations, and **(c–d)** stream fluxes from all scenarios. Stream water was the ~~overall~~ effluent from soil water and groundwater. The differences caused by hydrological differences (VF, HF, and PF) were much larger than the differences within each pair with the same flow partitioning, indicating significant hydrological impacts on weathering. ~~The Ca fluxes in the VF cases (100 % vertical flow) were higher than flow partitioning cases because they enabled maximum CO₂-charged water content with unweathered calcite at depth. Within each pair with the same flow pattern, the difference was mainly due to the distribution of soil CO₂. The difference in weathering fluxes was 1 %–12 % and 1 %–5 % between HF cases and VF cases, respectively, which is much smaller than differences between the PF cases. Comparing the HF and VF cases, differences in weathering fluxes were 73 % at 3.7 × 10⁻² m/a and 721 % at 3.7 m/a, which is about 1–2 orders of magnitude higher than differences induced by soil CO₂ distribution. Between the Grass_PF and Wood_PF cases, the differences were in the range of 17 %–207 % at the flow range of 3.7 × 10⁻²–3.7 m/a. The DIC fluxes showed similar trends (d).~~

than that in the grassland (Fig. 5c). Permeability evolution had a similar trend to porosity (not shown here). This indicates that if deeper roots promoted more water into deeper soils, they would push reaction fronts deeper and control the position where chemically unweathered bedrock was transformed into weathered bedrock. At timescales longer than century scale, calcite may become depleted, which ultimately reduces weathering rates (White and Brantley, 2003) and lead to similar weathering fronts in grasslands and woodlands.

## 4.3 The regulation of weathering rates by CO₂–carbonate contact

Natural systems are characterized by preferential flow paths such that flow distribution is not uniform in space. Weathering in such systems with preferential flow in zones of differing reactivities has been shown to hinge on the contact time between water and the reacting minerals instead of all minerals that are present (Wen and Li, 2018) (Eq. 1). Here

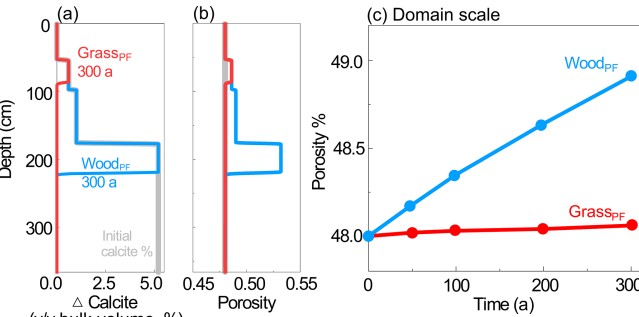

**Figure 5.** Predicted soil profiles of **(a)** calcite volume change (calcite = initial calcite volume − current calcite volume) and **(b)** porosity after 300 years; **(c)** predicted temporal evolution of domain-scale porosity in the grassland and woodland. The infiltration rate is 0.37 m/a. ~~Red and blue line represents the case Grass_PF and Wood_PF, respectively.~~

we contextualize the weathering rates in different scenarios (symbols) with predictions from an upscaled rate law developed by Wen and Li (2018) that incorporate the effects of heterogeneities in flow paths (Fig. 6). The time ratio $\frac{\tau_a}{\tau_{ad,r}}$ compares the domain water contact time (or residence time) with the contact time with dissolving calcite. Note that $\tau_a$ is total domain pore volume $V_T$/total water flow $Q_T$, and $\tau_{ad,r}$ is total reactive pore volume $V_r$/total water fluxes passing through reactive zone $Q_r$. Also note that water passing through the reactive zone stays in the subsurface longer, such that it is older water in general. The ratio $\frac{\tau_a}{\tau_{ad,r}}$ is therefore akin to the ~~older water fraction, i.e.,~~ the fraction of older water compared to the total water fraction. The older water fraction $F_{ow}$ is the counterpart of the young water fraction $F_{yw}$ ~~that has been~~ discussed in the literature (Kirchner, 2016, 2019). In Grass_VF and Wood_VF (open circles) where all CO₂-charged water flew through the deeper calcite zones, CO₂–calcite interactions reached maximum such that values of $\frac{\tau_a}{\tau_{ad,r}}$ approached 1.0, meaning all water interacted with calcite. Under this condition, weathering rates ~~are~~ the highest among all cases. In contrast, in Wood_HF and Grass_HF (open diamonds) where all CO₂-charged water bypassed the deeper calcite zone, $\frac{\tau_a}{\tau_{ad,r}}$ can be 1–3 orders of magnitude lower, and weathering rates were at their minima. The $\frac{\tau_a}{\tau_{ad,r}}$ value in the water partitioning cases (Grass_PF and Wood_PF, filled circles) fell in between. At the same $\tau_a$, the Wood_PF case with deepening roots promoted CO₂–water–calcite contact (i.e., larger $\frac{\tau_a}{\tau_{ad,r}}$) and dissolved calcite at higher rates.

The magnitude of the rate difference also depends on the overall flow rates (or domain contact time $\tau_a$). At fast flow with small $\tau_a$ (large symbols and thick lines in Fig. 6), flow partitioning has a larger influence. At $\tau_a$ of 0.47 years, the weathering rates in the VF cases were more than an order of magnitude higher than those in the PF cases. The weathering rate in the woodland was over 7 times that of the grassland. In contrast, at $\tau_a$ of 47 years, the rate differences between

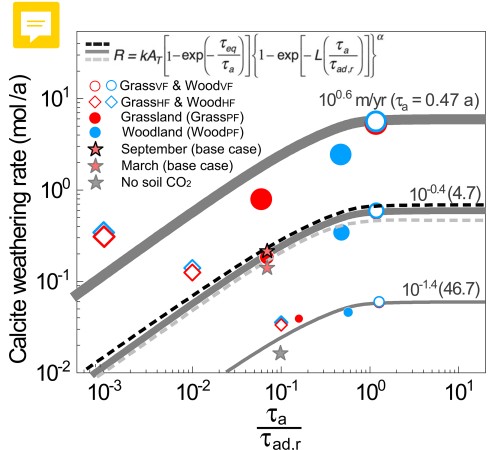

**Figure 6.** Calcite weathering rate as a function of the reactive time ratio $\frac{\tau_a}{\tau_{ad,r}}$, a proxy of the older, reactive water fraction compared to the total water fluxes. Symbols are rates from numerical experiments. Gray lines are predictions from the rate law equation at $\alpha = 0.8$ (Wen and Li, 2018). Large to small dots and thick to thin lines are for infiltration rates from $10^{0.6}$ to $10^{-1.4}$ m/a. The red filled stars represent the monthly rates in September (highest soil $CO_2$) and March (lowest soil $CO_2$) in the base case at $10^{-0.4}$ (0.37) m/a; the black to gray dashed lines represent predictions with increasing soil $CO_2$ level (i.e., larger $\tau_{eq}$) from Eq. (1). The gray filled star is for the case without soil $CO_2$. At any specific infiltration rate or $\tau_a$, the VF and HF cases bracket the two ends, whereas the PF cases fall in between. The Wood$_{PF}$ case with deeper roots enhanced the $CO_2$–water–calcite contact (i.e., larger $\frac{\tau_a}{\tau_{ad,r}}$ ratios), leading to higher calcite weathering rates compared to grasslands. The differences of weathering rates induced by different soil $CO_2$ level were relatively small compared to those hydrological changes induced by rooting depth.

grass and woody cases were less than 25 %, largely because the dissolution has reached equilibrium. In the VF and HF cases where flow conditions were the same, the soil $CO_2$ distribution in grassland and woodland differed only slightly, leading to similar values of $\frac{\tau_a}{\tau_{ad,r}}$ and weathering rates and indicating minimal impacts of soil $CO_2$ distribution. These rates from numerical experiments closely follow the prediction from the upscaled reaction rate law (gray lines, Eq. 1). The rate law predicted that weathering rates increased from HF cases with small $\frac{\tau_a}{\tau_{ad,r}}$ values to VF cases where $\frac{\tau_a}{\tau_{ad,r}}$ approached 1. It also showed that weathering rates reached their maxima in homogeneous domains without flow partitioning, or when the fraction of older, reactive water is essentially 1.0.

# 5 Discussion

Because carbonate dissolution is thermodynamically controlled and transport limited, the overall weathering rates depend on how much $CO_2$-charged water flushes through the carbonate zone. This work indicates that the roots potentially enhance weathering rates in two ways. First, roots can con-

trol thermodynamic limits of carbonate dissolution by regulating how much $CO_2$ is transported downward and enters the carbonate-rich zone. In fact, the base-case grassland data and model reveal that the concentrations of Ca and DIC are regulated by seasonal fluctuation of $pCO_2$ and soil respiration. Second, roots control how much and how frequently water fluxes through the carbonate zone export reaction products at equilibrium such that more dissolution can occur. The numerical experiments indicate that carbonate weathering at depth hinges on the rate of delivery of $CO_2$-enriched water. Deepening roots in woodlands that channel more water into the unweathered carbonate can elevate weathering rates by more than an order of magnitude compared to grasslands. Below we elaborate and discuss these two messages.

## 5.1 The thermodynamics of carbonate weathering: control of temperature and $pCO_2$

The base-case data and simulation showed that calcite dissolution reaches equilibrium rapidly and is thermodynamically controlled (Sect. 4.1), which echoes observations from other weathering studies (Tsypin and Macpherson, 2012; Gaillardet et al., 2019). The extent of dissolution, or solubility indicated in Ca and DIC concentrations, is determined by soil $CO_2$ levels that are a function of ecosystem functioning and climate. In hot, dry summer, soil respiration reaches its maximum rates in upper soil horizons and $pCO_2$ peaks (Fig. 2). In wet and cold winter, soil $pCO_2$ plummets, leading to much lower Ca and DIC concentrations. Based on Reactions (0–4) (Table 1) and temperature dependence of equilibrium constants, the following equations can be derived (detailed derivation in the Supplement) for Ca and DIC concentrations in carbonate-dominated landscapes:

$$C_{Ca} = \sqrt[3]{K_{t,25} \exp\left(-\frac{\Delta H_t^{\circ}}{4R} \times \left(\frac{1}{T} - \frac{1}{273.15 + 25}\right)\right) \cdot pCO_2},$$ 

(2)

$$C_{DIC} = 2\sqrt[3]{K_{t,25} \exp\left(-\frac{\Delta H_t^{\circ}}{4R} \times \left(\frac{1}{T} - \frac{1}{273.15 + 25}\right)\right) \cdot pCO_2}$$
$$+ K_{1,25} \exp\left(-\frac{\Delta H_1^{\circ}}{4R} \times \left(\frac{1}{T} - \frac{1}{273.15 + 25}\right)\right) \cdot pCO_2.$$

(3)

Here $K_{t,25}$ is the total equilibrium constant $K_t$ ($= \frac{a_{Ca2+} a_{HCO_3^-}^2}{pCO_2} = K_1 K_4$) of the combined Reactions (1) and (4) at 25 °C; $\Delta H_t^{\circ}$ is the corresponding standard enthalpy ($-35.83$ kJ/mol); $K_{1,25}$ and $\Delta H_1^{\circ}$ are the equilibrium constant and standard enthalpy of Reaction (1) (in Table 1); $R$ is the gas constant ($= 8.314 \times 10^{-3}$ kJ/K/mol). Equations (2) and (3) imply that DIC and Ca in upper soil water (0.2 m) are

lower compared to groundwater (3.6 m) in the base case at Konza, due to lower dissolved $CO_2$(aq) ~~arising from~~ higher temperature and higher diffusion rates in upper soil.

Equations (2) and (3) ~~were tested~~ with soil $pCO_2$ and spring water chemistry data from eight carbonate-dominated catchments (Dandurand et al., 1982; Lopez-Chicano et al., 2001; Moral et al., 2008; Ozkul et al., 2010; Kanduc et al., 2012; Calmels et al., 2014; Abongwa and Atekwana, 2015; Huang et al., 2015) (also see the Supplement for details). As shown in Fig. 7, spring water (~~as~~ representing groundwater) DIC and Ca concentrations increase with $pCO_2$~~, and~~ Eqs. (2)–(3) can describe these data and DIC and Ca levels from numerical simulations from this work (empty dots in Fig. 7). The lines of Eqs. (2)–(3) describe the relationship at 10 °C, the mean annual temperature, confirming the thermodynamics control of carbonate dissolution by soil $CO_2$ levels. The equation lines ~~closely~~ predicted Ca and DIC under high-$pCO_2$ and lower-pH conditions ($< 8.0$), because these conditions ensure the validity of the assumption of negligible $CO_3^{2-}$ in the derivation. The presence of cations and anions other than Ca and DIC can complicate the solution and can bring significant variations of DIC and Ca concentrations under the same $pCO_2$ conditions.

High temperature ($T = 17$ °C) leads to lower DIC and Ca concentrations by about 10 % due to the lower calcite and $CO_2$ solubility at higher $T$. Higher $T$ also elevates $pCO_2$ by enhancing soil respiration. Various equations ~~exist for predicting~~ soil $pCO_2$ based on climate and ecosystem functioning indicators such as net primary production (NPP) (Cerling, 1984; Goddéris et al., 2010; Romero-Mujalli et al., 2019a). These equations can be used together with Eqs. (2–3) for the estimation of Ca and DIC concentrations in carbonate-derived waters. Gaillardet et al. (2019) shows that $pCO_2$ can increase by 2 times with $T$ increasing from 9 to 17 °C, which can elevate DIC and Ca concentrations over 50 %. Macpherson et al. (2008) observed a 20 % increase in groundwater $pCO_2$ in Konza over a 15-year period and suggested that increased soil respiration ~~due to~~ climate ~~warming~~ may have elevated soil and groundwater $pCO_2$. Hasenmueller et al. (2015) demonstrated topographic controls on soil $CO_2$. ~~Variations of s~~oil $CO_2$ and Ca and DIC concentrations ~~therefore~~ are an integrated outcome of climate, soil respiration, subsurface structures, and hydrological conditions.

## 5.2 Hydrological controls of root-enhanced carbonate weathering

*Evidence from field data.* Data in Konza show that although soil $pCO_2$ peaks in summer, these peaks do not occur right away in deeper groundwater until about 2 months later. Macpherson et al. (2008) and Tsypin and Macpherson (2012) contributed ~~the delay~~ to the water travel time from the soil to the groundwater aquifer. The calculation of travel time based on depth difference in soil and groundwater sampling location (214 cm) and average velocity (0.37 m/a) indicates that

it will takes 7–8 months on average for water to reach deeper groundwater. The 2-month delay, much shorter than the estimated 7–8 months, suggests that $CO_2$-charged water may arrive at deeper zones via preferential flow facilitated by roots or other macropores~~. Macropores can take be~~ large conduits that ~~have been~~ observed in carbonate formations (Hartmann et al., 2014).

The numerical experiments suggest that the root-relevant hydrology can ~~play an~~ essential ~~role~~ in enhancing chemical weathering (Figs. 4 and 6). In general, the hydrological enhancement of weathering rates also alludes to the key connection between weathering and the transit (travel) time distribution (McGuire and McDonnell, 2006; Sprenger et al., 2019). In particular, water that routes through lateral paths is younger; water that penetrates deeper is older. Figure 6 says that weathering rates increase with increasing reactive water fraction, until reaching their maxima when all water is in contact with reactive minerals. This aligns with observations at Konza that woody-encroached watersheds exhibit higher Ca fluxes in streams and supports the hypothesis that deeper roots can enhance mineral–water interaction via deeper flow paths (Sullivan et al., 2019). Deepening roots can also enhance connectivity between shallow and deeper zones, therefore reducing concentration contrasts between soil water and groundwater. This can lead to more chemostatic $C–Q$ relationships as shown in $Wood_{PF}$ ($b = -0.14$ in $C = aQ^b$) compared to $Grass_{PF}$ ($b = -0.33$) (Fig. 4). These findings echo conclusions from Zhi et al. (2019) and Zhi and Li (2020) that chemistry differences in shallow versus deeper waters regulate $C–Q$ patterns. The $b$ values from the $C–Q$ relationships coming out of the 1-D modeling (Fig. 4) suggest that if flow partitioning is the only difference between the grassland and woody watersheds, a $C–Q$ relationship exhibiting dilution with negative $b$ values is expected in the grassland. The Konza stream data in Sullivan et al. (2019) however showed that $C–Q$ slopes in grasslands ($b = -0.003$) and in woody-encroached lands ($b = 0.013$) are both close to zero (Figs. 7 and 8 in Sullivan et al., 2019). The root influence on the $C–Q$ relationship therefore remains equivocal.

*Limitations of the model.* This discrepancy may suggest that ~~other~~ catchment features that are not represented in the simple 1-D model can influence $C–Q$ relationships. The model does not explicitly simulate how and to what degree root distribution at depth alters flow pathways. Instead we focus on the first-order principles of the hydrological ramification of roots. The numerical experiments took general observations of rooting characteristics in grasslands and woodlands and assumed that deepening roots in woodlands promote higher flow partitioning into the deep subsurface (Canadell et al., 1996; Nardini et al., 2016; Pawlik et al., 2016). In natural systems, however, other factors can also influence flow partitioning. ~~Some of these are not explicitly represented in our modeling exercises.~~ For example, contrasts in flow-conducting properties (i.e., porosity and per-

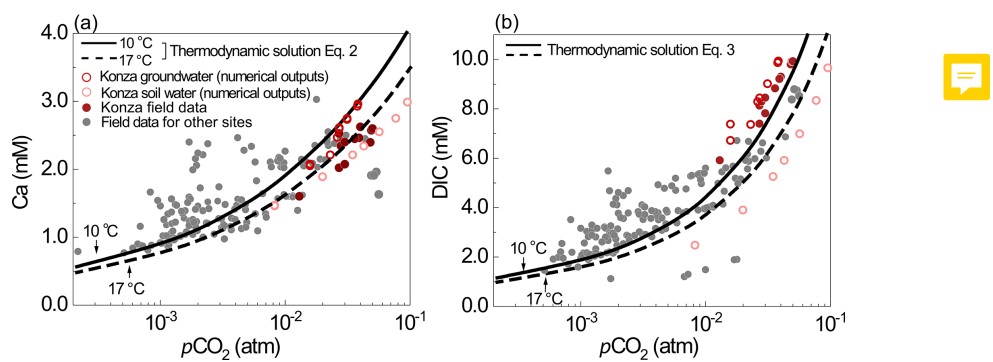

**Figure 7. (a)** Ca and **(b)** DIC data from the literature (gray dots) and prediction lines of Eqs. (2)–(3) at 10 (solid line) and 17 °C (dashed line). Measured spring water ~~chemistry is~~ from ~~calcite~~ dominant catchments in the literature: Abongwa and Atekwana (2015), Lopez-Chicano et al. (2001), Moral et al. (2008), Huang et al. (2015), Ozkul et al. (2010), Dandurand et al. (1982), Calmels et al. (2014), Kanduc et al. (2012), and Tsypin and Macpherson (2012).

meability) in shallow and deep zones, physical and chemical heterogeneity in carbonate distribution (Wen and Li, 2018), connectivity between different areas of the catchment in dry and wet times (Wen et al., 2020), and water table and corresponding lateral flow depth associated with the rainfall frequency and intensity (Li et al., 2017; Harman and Cosans, 2019). All these factors may affect the water ~~calcite~~ contact, leading to changes in ~~carbonate~~ weathering rates (Fig. 6). In addition to the alterations of hydrological flow paths, deepening roots may also affect the proportion of plant water uptake or soil water loss through transpiration (Pierret et al., 2016; Zhu et al., 2018), further modifying ~~water fluxes~~ into the deep subsurface layers. For example, studies ~~of semiarid woodlands~~ show that woody species predominantly use deep subsurface water while grasses predominantly use soil water from the upper soil layer (Ward et al., 2013). Furthermore, shallower root distributions in grasslands may be more efficient in using water from small rainfall events than forests with their deeper root distributions (Mazzacavallo and Kulmatiski, 2015). Other studies have ~~also~~ documented significant competition for water uptake in upper soil layers among both woody and grass species (Scholes and Archer, 1997). It remains inconclusive how these water uptake characteristics are best represented in numerical experiments. These processes are therefore not included at this point.

In addition, distributions of microbes and organic acids associated with rooting structures vary with sites and seasons, and root channels ~~and dry~~ conditions ~~with less soil water~~ may trigger calcite reprecipitation. Microbial activities surrounding living and dead roots can also lead to calcite precipitation and infilling of fractures and other macropores, as well as alter flow pathways (Lambers et al., 2009). Organic acids can decrease soil water pH, increase mineral solubilities through organic–metal complexations, and accelerate chemical weathering (Pittman and Lewan, 2012; Lawrence et al., 2014). Their impacts on carbonate weathering kinetics might be smaller (due to the fast kinetics) compared to their

alteration of calcite solubility in natural systems. Such processes further complicate how to represent biogeochemical processes in models. Although we do not explicitly simulate ~~all of~~ these ~~simultaneous and~~ competing processes ~~in our numerical experiments~~, the model was constrained by the field data in grasslands and woodland that have already integrated these effects in natural systems. Indeed, the weathering rates can be understood as the net weathering rates that are the net difference between dissolution and reprecipitation. This is reflected in lower carbonate weathering rates under low infiltration (low flow) conditions.

*The need for root-relevant measurements.* The data–model discrepancy highlights the limitation of a simple model but also points to the need for measurements. In fact, because carbonate weathering is transport limited and depends largely on water flow via carbonate-rich zones, colocated measurements of rooting characteristics, flow, and water chemistry at depth are essential. Root measurements however rarely go deeper than 30 cm (Richter and Billings, 2015). Existing work exploring root influence on weathering focused primarily on effects of soil $CO_2$ and root exudates (Drever, 1994; Lawrence et al., 2014; Gaillardet et al., 2019). Although it is well known that roots ~~play a paramount role~~ in modulating macropores and subsurface flow (Fan et al., 2017), the interactions between root characteristics, flow partitioning, and chemical weathering have remained poorly understood. Rooting characteristics depend on climate, plant species, topography, soil properties, and geology (Canadell et al., 1996; Mazvimavi et al., 2004; Price, 2011; Nardini et al., 2016). Further studies are needed to characterize root distribution beyond 30 cm, how they vary with intrinsic plant species and external conditions, and how and to what extent they alter subsurface flow. The first step could link rooting characteristics, including density and depth, to soil properties and borrow insights from existing relationships between soil properties and subsurface structure. For example, images of roots and pore structures can be used to characterize

the spatiotemporal heterogeneity of fluxes (Renard and Allard, 2013). Geostatistical indices such as permeability variance and correlation length ~~can~~ be used to quantify rooting structure and relate ~~them~~ [CE6] ~~f~~low partitioning and mineral weathering via numerical experiments (Wen and Li, 2017). The combination of numerical reactive transport experiments built on realistic rooting structure can help develop ~~a quantitative relationship between roots and flow partitioning and could support~~ models for estimating the influence of rooting dynamics on water and carbon cycles at the catchment scale.

### 5.3 Deepening roots enhance carbon fluxes into the deep subsurface: a potential carbon sink?

Mounting evidence has shown that the terrestrial system has become a stronger carbon sink in recent decades (Heimann and Reichstein, 2008), potentially accounting for the missing carbon sink as large as $\sim 1$ Pg C/a in the global carbon budget (Houghton, 2007; Cole et al., 2007). Although still much debated, recent studies have proposed the downward transport of soil-respired $CO_2$ and DIC into groundwater aquifers in deserts as a possible carbon sink in the global carbon cycle (Ma et al., 2014; Li et al., 2015). Considering the longer residence time of DIC in groundwater ($10^2$–$10^4$ years) than in the atmosphere ($\sim 10^1$–$10^2$ years; Archer and Brovkin, 2008), groundwater may act as a $CO_2$ storage sink. This work indicated that deepening roots can potentially reroute DIC fluxes to deeper groundwater storage. In particular, vegetation in dry places like deserts often has deep roots (Gupta et al., 2020). In fact, plants are known for growing deeper roots to tap groundwater during droughts (Brunner et al., 2015). Deepening roots can enhance downward water drainage (i.e., high vertical connectivity) to the depth and potentially facilitate the transport of DIC fluxes into the deep subsurface. As the pace of climate change accelerates, summer droughts are expected to intensify, which can potentially channel more $CO_2$ into the deeper subsurface via deepening roots.

With constraints from soil $CO_2$ data, the simulated $CO_2$ production rates in Grass$_{PF}$ and Wood$_{PF}$ are similarly at $\sim 0.6$ mol C/m$^2$/a under the infiltration rate of 0.37 m/a (Fig. S2). This is at the low end of the belowground net production (NPP) estimations at the Konza site ($\sim 2.0$–30.0 mol C/m$^2$/a), assuming that belowground NPP accounts for 50 % of the total NPP (Lett et al., 2004; Knapp and Ojima, 2014). The simulations showed that in woodlands the DIC downward fluxes can be $> 2.0$ times higher than those in grasslands (Fig. 4). In other words, more soil-respired $CO_2$ can transport to groundwater and become stored there for centuries to millennia before entering a stream. At the short timescale, this enhanced downward transport will reduce $CO_2$ escape back into the atmosphere. This may explain the observations at Konza that woody encroachment increased NPP; however, soil $CO_2$ flux was significantly reduced compared to the open grassland (Lett et al., 2004). Changing land cover (e.g., woody encroachment and boreal

forest creep) however is not the only mechanism for a carbon sink (Stevens et al., 2017; Wang et al., 2020). Older aged forests also tend to have deepening roots and may act as the carbon sink (Luyssaert et al., 2008), although this is not the case in Konza.

## 6 Conclusions

This work aims to understand thermodynamic and hydrological control of carbonate weathering driven by deepening roots. Field data and reactive transport simulation for a grassland in the Konza Prairie LTER site suggest that carbonate dissolution is thermodynamically controlled, and seasonal changes in temperature and $pCO_2$ drive variations of Ca and DIC concentrations in soil water and groundwater. We derived equations based on reaction thermodynamics (Eqs. 2–3) to estimate Ca and DIC as a function of $pCO_2$ and temperature, which have been shown to be applicable in other carbonate-dominated systems. The numerical experiments probed the potential effects of deepening roots on weathering by channeling a higher proportion of vertically downward flow (40 % of the total) into the deep subsurface with abundant calcite. The results show that deeper penetration of roots and higher vertical flow (recharge) enhanced $CO_2$–carbonate contact. At an infiltration rate of 3.7 m/a, calcite weathering flux in woodlands was ~~207~~ % higher than that in grasslands. At $3.7 \times 10^{-2}$ m/a, the weathering flux in woodland was 17 % higher. The hydrological impacts on carbonate weathering were ~~more than 10 times~~ higher than the biogeochemical impacts via elevated soil $CO_2$ alone, underscoring the importance of ~~root presence~~ in weathering. The modeling demonstrates that ~~the~~ weathering rates depend on flow partitioning (the older water fraction that penetrates deeper) and relative magnitude of water contact time in the deep carbonate zone. At the century scale, with the higher proportion of vertical flow, the deeper roots pushed the weathering fronts 2 times deeper and resulted in a 10-times greater increase in porosity and permeability. Broadly, this deeper propagation of reaction fronts may accelerate rates of channel incision and hillslope erosion (Lebedeva and Brantley, 2013; Brantley et al., 2017b) and therefore speed up landscape evolution (Phillips, 2005). It alludes to the importance of considering changes in subsurface hydrological flows associated with shifts in vegetation types and ~~measuring~~ rooting characteristics. This is particularly relevant as we assess the effects of climate change, land cover, and elevated atmosphere $CO_2$ concentrations on chemical weathering and carbon cycles.

*Data availability.* All data used for model parameterization can be acquired from http://lter.konza.ksu.edu/data [TS9]. The input files necessary to reproduce the results are available from the authors upon request (https://iee.psu.edu/content/li-li [TS10 TS11]).

*Supplement.* The supplement related to this article is available online at: https://doi.org/10.5194/bg-17-1-2020-supplement. TS12

*Author contributions.* HW, PS, and LL initiated the idea and designed the numerical experiments. GLM and SB provided the field data. HW ran the simulations, analyzed simulation results, and wrote the first draft of the manuscript. All coauthors participated in editing the manuscript.

*Competing interests.* The authors declare that they have no conflict of interest.

*Acknowledgements.* ~~We acknowledge funding support from the Konza Prairie LTER program (NSF DEB 1440484), the Signals in the Soils (SitS, EAR 1331726), and the NSF project (EAR 1911960).~~ We appreciate the assistance of the Konza Prairie Biological Station. ~~We are grateful for~~ the field data provided by Misha Tsypin and Zachary Brecheisen.

*Financial support.* This research has been supported by the National Science Foundation (grant nos. DEB-1440484, EAR–1331726, and EAR-1911960). TS13

*Review statement.* This paper was edited by Yakov Kuzyakov and reviewed by five anonymous referees.

# References 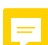

Abongwa, P. T. and Atekwana, E. A.: Controls on the chemical and isotopic composition of carbonate springs during evolution to saturation with respect to calcite, Chem. Geol., 404, 136–149, https://doi.org/10.1016/j.chemgeo.2015.03.024, 2015.

Ahrens, B., Braakhekke, M. C., Guggenberger, G., Schrumpf, M., and Reichstein, M.: Contribution of sorption, DOC transport and microbial interactions to the 14C age of a soil organic carbon profile: Insights from a calibrated process model, Soil Biol. Biochem., 88, 390–402, https://doi.org/10.1016/j.soilbio.2015.06.008, 2015.

Andrews, J. A. and Schlesinger, W. H.: Soil $CO_2$ dynamics, acidification, and chemical weathering in a temperate forest with experimental $CO_2$ enrichment, Global Biogeochem. Cy., 15, 149–162, https://doi.org/10.1029/2000gb001278, 2001.

Angers, D. A. and Caron, J.: Plant-induced changes in soil structure: Processes and feedbacks, Biogeochemistry, 42, 55–72, https://doi.org/10.1023/a:1005944025343, 1998.

Aragão, L. E. O. C., Malhi, Y., Metcalfe, D. B., Silva-Espejo, J. E., Jiménez, E., Navarrete, D., Almeida, S., Costa, A. C. L., Salinas, N., Phillips, O. L., Anderson, L. O., Alvarez, E., Baker, T. R., Goncalvez, P. H., Huamán-Ovalle, J., Mamani-Solórzano, M., Meir, P., Monteagudo, A., Patiño, S., Peñuela, M. C., Prieto, A., Quesada, C. A., Rozas-Dávila, A., Rudas, A., Silva Jr., J. A., and Vásquez, R.: Above- and below-ground net primary productivity across ten Amazonian forests on contrasting soils, Biogeosciences, 6, 2759–2778, https://doi.org/10.5194/bg-6-2759-2009, 2009.

Archer, D. and Brovkin, V.: The millennial atmospheric lifetime of anthropogenic $CO_2$, Clim. Change, 90, 283–297, https://doi.org/10.1007/s10584-008-9413-1, 2008.

Beerling, D. J., Chaloner, W. G., Woodward, F. I., and Berner, R. A.: The carbon cycle and carbon dioxide over Phanerozoic time: the role of land plants, Philos. T. Roy. Soc. B, 353, 75–82, https://doi.org/10.1098/rstb.1998.0192, 1998.

Bengtson, P. and Bengtsson, G.: Rapid turnover of DOC in temperate forests accounts for increased $CO_2$ production at elevated temperatures, Ecol. Lett., 10, 783–790, https://doi.org/10.1111/j.1461-0248.2007.01072.x, 2007.

Berner, E. K. and Berner, R. A.: Global environment: water, air, and geochemical cycles, Princeton University Press, Princeton, USA, 2012.

Berner, R. A.: Weathering, plants, and the long-term carbon cycle, Geochim. Cosmochim. Ac., 56, 3225–3231, https://doi.org/10.1016/0016-7037(92)90300-8, 1992.

Berner, R. A.: Paleoclimate – The rise of plants and their effect on weathering and atmospheric $CO_2$, Science, 276, 544–546, https://doi.org/10.1126/science.276.5312.544, 1997.

Beven, K. and Germann, P.: Macropores and water flow in soils revisited, Water Resour. Res., 49, 3071–3092, https://doi.org/10.1002/wrcr.20156, 2013.

Billings, S. A., Hirmas, D., Sullivan, P. L., Lehmeier, C. A., Bagchi, S., Min, K., Brecheisen, Z., Hauser, E., Stair, R., Flournoy, R., and Richter, D. D.: Loss of deep roots limits biogenic agents of soil development that are only partially restored by decades of forest regeneration, Elem. Sci. Anth., 6, 34–53 , https://doi.org/10.1525/elementa.287, 2018.

Brantley, S. L., Eissenstat, D. M., Marshall, J. A., Godsey, S. E., Balogh-Brunstad, Z., Karwan, D. L., Papuga, S. A., Roering, J., Dawson, T. E., Evaristo, J., Chadwick, O., McDonnell, J. J., and Weathers, K. C.: Reviews and syntheses: on the roles trees play in building and plumbing the critical zone, Biogeosciences, 14, 5115–5142, https://doi.org/10.5194/bg-14-5115-2017, 2017a.

Brantley, S. L., Lebedeva, M. I., Balashov, V. N., Singha, K., Sullivan, P. L., and Stinchcomb, G.: Toward a conceptual model relating chemical reaction fronts to water flow paths in hills, Geomorphology, 277, 100–117, https://doi.org/10.1016/j.geomorph.2016.09.027, 2017b.

Breecker, D. O., Sharp, Z. D., and McFadden, L. D.: Atmospheric $CO_2$ concentrations during ancient greenhouse climates were similar to those predicted for AD 2100, P. Natl. Acad. Sci. USA, 107, 576–580, https://doi.org/10.1073/pnas.0902323106, 2010.

Brunner, I., Herzog, C., Dawes, M. A., Arend, M., and Sperisen, C.: How tree roots respond to drought, Front. Plant Sci., 6, 547, doi:10.3389/fpls.2015.00547, 2015.

Calmels, D., Gaillardet, J., and Francois, L.: Sensitivity of carbonate weathering to soil $CO_2$ production by biological activity along a temperate climate transect, Chem. Geol., 390, 74–86, https://doi.org/10.1016/j.chemgeo.2014.10.010, 2014.

Canadell, J., Jackson, R. B., Ehleringer, J. R., Mooney, H. A., Sala, O. E., and Schulze, E. D.: Maximum rooting depth of vegetation types at the global scale, Oecologia, 108, 583–595, https://doi.org/10.1007/bf00329030, 1996.

Carey, J. C., Tang, J. W., Templer, P. H., Kroeger, K. D., Crowther, T. W., Burton, A. J., Dukes, J. S., Emmett, B., Frey, S. D., Heskel, M. A., Jiang, L., Machmuller, M. B., Mohan, J., Panetta, A. M., Reich, P. B., Reinsch, S., Wang, X., Allison, S. D., Bamminger, C., Bridgham, S., Collins, S. L., De Dato, G., Eddy, W. C., Enquist, B. J., Estiarte, M., Harte, J., Henderson, A., Johnson, B. R., Larsen, K. S., Luo, Y., Marhan, S., Melillo, J. M., Peuelas, J., Pfeifer-Meister, L., Poll, C., Rastetter, E., Reinmann, A. B., Reynolds, L. L., Schmidt, I. K., Shaver, G. R., Strong, A. L., Suseela, V., and Tietema, A.: Temperature response of soil respiration largely unaltered with experimental warming, P. Natl. Acad. Sci. USA, 113, 13797–13802, https://doi.org/10.1073/pnas.160536511, 2016.

Cerling, T. E.: The stable isotopic composition of modern soil carbonate and its relationship to climate, Earth Planet. Sc. Lett., 71, 229–240, https://doi.org/10.1016/0012-821x(84)90089-x, 1984.

Cheng, J. H., Zhang, H. J., Wang, W., Zhang, Y. Y., and Chen, Y. Z.: Changes in Preferential Flow Path Distribution and Its Affecting Factors in Southwest China, Soil Sci., 176, 652–660, https://doi.org/10.1097/SS.0b013e31823554ef, 2011.

Cole, J. J., Prairie, Y. T., Caraco, N. F., McDowell, W. H., Tranvik, L. J., Striegl, R. G., Duarte, C. M., Kortelainen, P., Downing, J. A., Middelburg, J. J., and Melack, J.: Plumbing the global carbon cycle: Integrating inland waters into the terrestrial carbon budget, Ecosystems, 10, 171–184, https://doi.org/10.1007/s10021-006-9013-8, 2007.

Costa, A.: Permeability relationship: A reexamination of the Kozeny equation based on a fractal pore geometry assumption, Geophys. Res. Lett., 33, L02318, https://doi.org/10.1029/2005GL025134, 2006.

Covington, M. D., Gulley, J. D., and Gabrovsek, F.: Natural variations in calcite dissolution rates in streams: Controls, implications, and open questions, Geophys. Res. Lett., 42, 2836–2843, https://doi.org/10.1002/2015gl063044, 2015.

Dandurand, J. L., Gout, R., Hoefs, J., Menschel, G., Schott, J., and Usdowski, E.: Kinetically controlled variations of major components and carbon and oxygen isotopes in a calcite-precipitating spring, Chem. Geol., 36, 299–315, https://doi.org/10.1016/0009-2541(82)90053-5, 1982.

Deng, H., Voltolini, M., Molins, S., Steefel, C., DePaolo, D., Ajo-Franklin, J., and Yang, L.: Alteration and Erosion of Rock Matrix Bordering a Carbonate-Rich Shale Fracture, Environ. Sci. Tech., 51, 8861–8868, https://doi.org/10.1021/acs.est.7b02063, 2017.

Drever, J. I.: The effect of land plants on weathering rates of silicate minerals, Geochim. Cosmochim. Ac., 58, 2325–2332, https://doi.org/10.1016/0016-7037(94)90013-2, 1994.

Eberbach, P. L.: The eco-hydrology of partly cleared, native ecosystems in southern Australia: a review, Plant Soil, 257, 357–369, https://doi.org/10.1023/A:1027392703312, 2003.

Fan, Y., Miguez-Macho, G., Jobbagy, E. G., Jackson, R. B., and Otero-Casal, C.: Hydrologic regulation of plant rooting depth, P. Natl. Acad. Sci. USA, 114, 10572–10577, https://doi.org/10.1073/pnas.1712381114, 2017.

Frank, D. A., Pontes, A. W., Maine, E. M., Caruana, J., Raina, R., Raina, S., and Fridley, J. D.: Grassland root communities: species distributions and how they are linked to aboveground abundance, Ecology, 91, 3201–3209, https://doi.org/10.1890/09-1831.1, 2010.

Gaillardet, J., Calmels, D., Romero-Mujalli, G., Zakharova, E., and Hartmann, J.: Global climate control on carbonate weathering intensity, Chem. Geol., 527, 118762, https://doi.org/10.1016/j.chemgeo.2018.05.009, 2019.

Gill, R. A., Kelly, R. H., Parton, W. J., Day, K. A., Jackson, R. B., Morgan, J. A., Scurlock, J. M. O., Tieszen, L. L., Castle, J. V., Ojima, D. S., and Zhang, X. S.: Using simple environmental variables to estimate below-ground productivity in grasslands, Global Ecol. Biogeogr., 11, 79–86, https://doi.org/10.1046/j.1466-822X.2001.00267.x, 2002.

Goddéris, Y., Williams, J. Z., Schott, J., Pollard, D., and Brantley, S. L.: Time evolution of the mineralogical composition of Mississippi Valley loess over the last 10 kyr: Climate and geochemical modeling, Geochim. Cosmochim. Ac., 74, 6357–6374, https://doi.org/10.1016/j.gca.2010.08.023, 2010.

Gupta, A., Rico-Medina, A., and Cano-Delgado, A. I.: The physiology of plant responses to drought, Science, 368, 266–269, https://doi.org/10.1126/science.aaz7614, 2020.

Harman, C. J. and Cosans, C. L.: A low-dimensional model of bedrock weathering and lateral flow coevolution in hillslopes: 2. Controls on weathering and permeability profiles, drainage hydraulics, and solute export pathways, Hydrol. Process., 33, 1168–1190, https://doi.org/10.1002/hyp.13385, 2019.

Harper, C. W., Blair, J. M., Fay, P. A., Knapp, A. K., and Carlisle, J. D.: Increased rainfall variability and reduced rainfall amount decreases soil $CO_2$ flux in a grassland ecosystem, Global Change Biol., 11, 322–334, https://doi.org/10.1111/j.1365-2486.2005.00899.x, 2005.

Hartmann, A., Goldscheider, N., Wagener, T., Lange, J., and Weiler, M.: Karst water resources in a changing world: Review of hydrological modeling approaches, Rev. Geophys., 52, 218–242, https://doi.org/10.1002/2013RG000443, 2014.

Hasenmueller, E. A., Jin, L. X., Stinchcomb, G. E., Lin, H., Brantley, S. L., and Kaye, J. P.: Topographic controls on the depth distribution of soil $CO_2$ in a small temperate watershed, Appl. Geochem., 63, 58–69, https://doi.org/10.1016/j.apgeochem.2015.07.005, 2015.

Hasenmueller, E. A., Gu, X., Weitzman, J. N., Adams, T. S., Stinchcomb, G. E., Eissenstat, D. M., Drohan, P. J., Brantley, S. L., and Kaye, J. P.: Weathering of rock to regolith: The activity of deep roots in bedrock fractures, Geoderma, 300, 11–31, doi:/10.1016/j.geoderma.2017.03.020, 2017.

Hauser, E., Richter, D. D., Markewitz, D., Brecheisen, Z., and Billings, S. A.: Persistent anthropogenic legacies structure depth dependence of regenerating rooting systems and their functions, Biogeochemistry, 147, 259–275, https://doi.org/10.1007/s10533-020-00641-2, 2020.

Heimann, M. and Reichstein, M.: Terrestrial ecosystem carbon dynamics and climate feedbacks, Nature, 451, 289–292, https://doi.org/10.1038/nature06591, 2008.

Houghton, R. A.: Balancing the global carbon budget, Annu. Rev. Earth Pl. Sc., 35, 313–347, https://doi.org/10.1146/annurev.earth.35.031306.140057, 2007.

Huang, F., Zhang, C. L., Xie, Y. C., Li, L., and Cao, J. H.: Inorganic carbon flux and its source in the karst catchment of Maocun, Guilin, China, Environ, Earth Sci, 74, 1079–1089, https://doi.org/10.1007/s12665-015-4478-4, 2015.

Jackson, R. B., Canadell, J., Ehleringer, J. R., Mooney, H. A., Sala, O. E., and Schulze, E. D.: A global analysis of root

distributions for terrestrial biomes, Oecologia, 108, 389–411, https://doi.org/10.1007/bf00333714, 1996.

Jiang, H., Deng, Q., Zhou, G., Hui, D., Zhang, D., Liu, S., Chu, G., and Li, J.: Responses of soil respiration and its temperature/moisture sensitivity to precipitation in three subtropical forests in southern China, Biogeosciences, 10, 3963–3982, https://doi.org/10.5194/bg-10-3963-2013, 2013.

Jobbagy, E. G. and Jackson, R. B.: The vertical distribution of soil organic carbon and its relation to climate and vegetation, Ecol. Appl., 10, 423–436, https://doi.org/10.2307/2641104, 2000.

Kanduc, T., Mori, N., Kocman, D., Stibilj, V., and Grassa, F.: Hydrogeochemistry of Alpine springs from North Slovenia: Insights from stable isotopes, Chem. Geol., 300, 40–54, https://doi.org/10.1016/j.chemgeo.2012.01.012, 2012.

Kirchner, J. W.: Aggregation in environmental systems – Part 1: Seasonal tracer cycles quantify young water fractions, but not mean transit times, in spatially heterogeneous catchments, Hydrol. Earth Syst. Sci., 20, 279–297, https://doi.org/10.5194/hess-20-279-2016, 2016.

Kirchner, J. W.: Quantifying new water fractions and transit time distributions using ensemble hydrograph separation: theory and benchmark tests, Hydrol. Earth Syst. Sci., 23, 303–349, https://doi.org/10.5194/hess-23-303-2019, 2019.

Knapp, A. and Ojima, D.: NPP Grassland: Konza Prairie, USA, 1984-1990, R1, ORNL DAAC, 2014. TS14

Kozeny, J.: Über kapillare Leitung des Wassers im Boden, Royal Academy of Science, Vienna, 136, 271–306, 1927.

Lambers, H., Mougel, C., Jaillard, B., and Hinsinger, P.: Plant-microbe-soil interactions in the rhizosphere: an evolutionary perspective, Plant Soil, 321, 83–115, https://doi.org/10.1007/s11104-009-0042-x, 2009.

Lawrence, C., Harden, J., and Maher, K.: Modeling the influence of organic acids on soil weathering, Geochim. Cosmochim. Ac., 139, 487–507, https://doi.org/10.1016/j.gca.2014.05.003, 2014.

Lebedeva, M. I. and Brantley, S. L.: Exploring geochemical controls on weathering and erosion of convex hillslopes: beyond the empirical regolith production function, Earth Surf. Proc. Land., 38, 1793–1807, https://doi.org/10.1002/esp.3424, 2013.

Lett, M. S., Knapp, A. K., Briggs, J. M., and Blair, J. M.: Influence of shrub encroachment on aboveground net primary productivity and carbon and nitrogen pools in a mesic grassland, Can. J. Bot., 82, 1363–1370, https://doi.org/10.1139/b04-088, 2004.

Li, L., Bao, C., Sullivan, P. L., Brantley, S., Shi, Y. N., and Duffy, C.: Understanding watershed hydrogeochemistry: 2. Synchronized hydrological and geochemical processes drive stream chemostatic behavior, Water Resour. Res., 53, 2346–2367, https://doi.org/10.1002/2016wr018935, 2017.

Li, Y., Wang, Y. G., Houghton, R. A., and Tang, L. S.: Hidden carbon sink beneath desert, Geophys. Res. Lett., 42, 5880–5887, https://doi.org/10.1002/2015gl064222, 2015.

Lopez-Chicano, M., Bouamama, M., Vallejos, A., and Pulido-Bosch, A.: Factors which determine the hydrogeochemical behaviour of karstic springs. A case study from the Betic Cordilleras, Spain, Appl. Geochem., 16, 1179–1192, https://doi.org/10.1016/s0883-2927(01)00012-9, 2001.

Luyssaert, S., Schulze, E. D., Borner, A., Knohl, A., Hessenmoller, D., Law, B. E., Ciais, P., and Grace, J.: Old-growth forests as global carbon sinks, Nature, 455, 213–215, https://doi.org/10.1038/nature07276, 2008.

Ma, J., Liu, R., Tang, L.-S., Lan, Z.-D., and Li, Y.: A downward $CO_2$ flux seems to have nowhere to go, Biogeosciences, 11, 6251–6262, https://doi.org/10.5194/bg-11-6251-2014, 2014.

Macpherson, G. L.: Hydrogeology of thin limestones: The Konza Prairie Long-Term Ecological Research Site, Northeastern Kansas, J. Hydrol., 186, 191–228, https://doi.org/10.1016/s0022-1694(96)03029-6, 1996.

Macpherson, G. L. and Sullivan, P. L. J. C. G.: Dust, impure calcite, and phytoliths: Modeled alternative sources of chemical weathering solutes in shallow groundwater, Chem. Geol., 527, 118871, https://doi.org/10.1016/j.chemgeo.2018.08.007, 2019.

Macpherson, G. L., Roberts, J. A., Blair, J. M., Townsend, M. A., Fowle, D. A., and Beisner, K. R.: Increasing shallow groundwater $CO_2$ and limestone weathering, Konza Prairie, USA, Geochim. Cosmochim. Ac., 72, 5581–5599, https://doi.org/10.1016/j.gca.2008.09.004, 2008.

Mazvimavi, D., Meijerink, A. M. J., and Stein, A.: Prediction of base flows from basin characteristics: a case study from Zimbabwe, Hydrolog. Sci. J., 49, 703–715, https://doi.org/10.1623/hysj.49.4.703.54428, 2004.

Mazzacavallo, M. G. and Kulmatiski, A.: Modelling Water Uptake Provides a New Perspective on Grass and Tree Coexistence, Plos One, 10, e0144300, https://doi.org/10.1371/journal.pone.0144300, 2015.

McGuire, K. J. and McDonnell, J. J.: A review and evaluation of catchment transit time modeling, J. Hydrol., 330, 543–563, https://doi.org/10.1016/j.jhydrol.2006.04.020, 2006.

Moral, F., Cruz-Sanjulian, J. J., and Olias, M.: Geochemical evolution of groundwater in the carbonate aquifers of Sierra de Segura (Betic Cordillera, southern Spain), J. Hydrol., 360, 281–296, https://doi.org/10.1016/j.jhydrol.2008.07.012, 2008.

Moriasi, D. N., Arnold, J. G., Van Liew, M. W., Bingner, R. L., Harmel, R. D., and Veith, T. L.: Model evaluation guidelines for systematic quantification of accuracy in watershed simulations, T. Asabe, 50, 885–900, https://doi.org/10.13031/2013.23153, 2007.

Mottershead, D. N., Baily, B., Collier, P., and Inkpen, R. J.: Identification and quantification of weathering by plant roots, Build. Environ., 38, 1235–1241, https://doi.org/10.1016/s0360-1323(03)00080-5, 2003.

Nardini, A., Casolo, V., Dal Borgo, A., Savi, T., Stenni, B., Bertoncin, P., Zini, L., and McDowell, N. G.: Rooting depth, water relations and non-structural carbohydrate dynamics in three woody angiosperms differentially affected by an extreme summer drought, Plant Cell Environ., 39, 618–627, https://doi.org/10.1111/pce.12646, 2016.

Neff, J. C. and Hooper, D. U.: Vegetation and climate controls on potential $CO_2$, DOC and DON production in northern latitude soils, Glob. Change Biol., 8, 872–884, https://doi.org/10.1046/j.1365-2486.2002.00517.x, 2002.

Nepstad, D. C., Decarvalho, C. R., Davidson, E. A., Jipp, P. H., Lefebvre, P. A., Negreiros, G. H., Dasilva, E. D., Stone, T. A., Trumbore, S. E., and Vieira, S.: The role of deep roots in the hydrological and carbon cycles of amazonian forests and pastures, Nature, 372, 666–669, https://doi.org/10.1038/372666a0, 1994.

Nippert, J. B., Wieme, R. A., Ocheltree, T. W., and Craine, J. M.: Root characteristics of $C_4$ grasses limit reliance on deep soil water in tallgrass prairie, Plant Soil, 355, 385–394, https://doi.org/10.1007/s11104-011-1112-4, 2012.

Please note the remarks at the end of the manuscript.

Noguchi, S., Tsuboyama, Y., Sidle, R. C., and Hosoda, I.: Spatially distributed morphological characteristics of macropores in forest soils of Hitachi Ohta Experimental Watershed, Japan, J. For. Res.-Jpn., 2, 207–215, https://doi.org/10.1007/BF02348317, 1997.

Oades, J. M.: The role of biology in the formation, stabilization and degradation of soil structure, Geoderma, 56, 377–400, https://doi.org/10.1016/0016-7061(93)90123-3, 1993.

Ozkul, M., Gokgoz, A., and Horvatincic, N.: Depositional properties and geochemistry of Holocene perched springline tufa deposits and associated spring waters: a case study from the Denizli Province, Western Turkey, in: Tufas and Speleothems: Unravelling the Microbial and Physical Controls, edited by: Pedley, H. M. and Rogerson, M., Geological Society Special Publication, London, UK, 245–262, 2010.

Palandri, J. L. and Kharaka, Y. K.: A compilation of rate parameters of water-mineral interaction kinetics for application to geochemical modeling, U. S. Geological Survey, Menlo Park CA, USA, 70 pp., 2004.

Patrick, L., Cable, J., Potts, D., Ignace, D., Barron-Gafford, G., Griffith, A., Alpert, H., Van Gestel, N., Robertson, T., Huxman, T. E., Zak, J., Loik, M. E., and Tissue, D.: Effects of an increase in summer precipitation on leaf, soil, and ecosystem fluxes of $CO_2$ and $H_2O$ in a sotol grassland in Big Bend National Park, Texas, Oecologia, 151, 704–718, https://doi.org/10.1007/s00442-006-0621-y, 2007.

Pawlik, L., Phillips, J. D., and Samonil, P.: Roots, rock, and regolith: Biomechanical and biochemical weathering by trees and its impact on hillslopes – A critical literature review, Earth-Sci. Rev., 159, 142–159, https://doi.org/10.1016/j.earscirev.2016.06.002, 2016.

Pennell, K. D., Boyd, S. A., and Abriola, L. M.: Surface-area of soil organic-matter reexamined, Soil Sci. Soc. Am. J., 59, 1012–1018, https://doi.org/10.2136/sssaj1995.03615995005900040008x, 1995.

Phillips, J. D.: Weathering instability and landscape evolution, Geomorphology, 67, 255–272, https://doi.org/10.1016/j.geomorph.2004.06.012, 2005.

Pierret, A., Maeght, J. L., Clement, C., Montoroi, J. P., Hartmann, C., and Gonkhamdee, S.: Understanding deep roots and their functions in ecosystems: an advocacy for more unconventional research, Ann. Bot.-London, 118, 621–635, https://doi.org/10.1093/aob/mcw130, 2016.

Pittman, E. D. and Lewan, M. D.: Organic acids in geological processes, Springer Science Business Media, Berlin, Germany, 2012.

Plummer, L., Wigley, T., and Parkhurst, D.: The kinetics of calcite dissolution in $CO_2$-water systems at 5 degrees to 60 degrees C and 0.0 to 1.0 atm $CO_2$, Am, J, Sci,, 278, 179–216, https://doi.org/10.2475/ajs.278.2.179, 1978.

Price, K.: Effects of watershed topography, soils, land use, and climate on baseflow hydrology in humid regions: A review, Prog. Phys. Geog., 35, 465–492, https://doi.org/10.1177/0309133311402714, 2011.

Renard, P. and Allard, D.: Connectivity metrics for subsurface flow and transport, Adv. Water Resour., 51, 168–196, https://doi.org/10.1016/j.advwatres.2011.12.001, 2013.

Richter, D. D. and Billings, S. A.: 'One physical system': Tansley's ecosystem as Earth's critical zone, New Phytol., 206, 900–912, https://doi.org/10.1111/nph.13338, 2015.

Romero-Mujalli, G., Hartmann, J., and Börker, J.: Temperature and $CO_2$ dependency of global carbonate weathering fluxes – Implications for future carbonate weathering research, Chem. Geol., 527, 118874, https://doi.org/10.1016/j.chemgeo.2018.08.010, 2019a.

Romero-Mujalli, G., Hartmann, J., Börker, J., Gaillardet, J., and Calmels, D.: Ecosystem controlled soil-rock $pCO_2$ and carbonate weathering – Constraints by temperature and soil water content, Chem. Geol., 527, 118634, https://doi.org/10.1016/j.chemgeo.2018.01.030, 2019b.

Sadras, V. O.: Influence of size of rainfall events on water-driven processes I. Water budget of wheat crops in southeastern Australia, Aust. J. Agric. Res., 54, 341–351, https://doi.org/10.1071/ar02112, 2003.

Scholes, R. J. and Archer, S. R.: Tree-grass interactions in savannas, Annu. Rev. Ecol. Syst., 28, 517–544, https://doi.org/10.1146/annurev.ecolsys.28.1.517, 1997.

Sprenger, M., Stumpp, C., Weiler, M., Aeschbach, W., Allen, S. T., Benettin, P., Dubbert, M., Hartmann, A., Hrachowitz, M., Kirchner, J. W., McDonnell, J. J., Orlowski, N., Penna, D., Pfahl, S., Rinderer, M., Rodriguez, N., Schmidt, M., and Werner, C.: The Demographics of Water: A Review of Water Ages in the Critical Zone, Rev. Geophys., 57, 800–834, https://doi.org/10.1029/2018rg000633, 2019.

Steefel, C. I., Appelo, C. A. J., Arora, B., Jacques, D., Kalbacher, T., Kolditz, O., Lagneau, V., Lichtner, P. C., Mayer, K. U., Meeussen, J. C. L., Molins, S., Moulton, D., Shao, H., Šimůnek, J., Spycher, N., Yabusaki, S. B., and Yeh, G. T.: Reactive transport codes for subsurface environmental simulation, Computat. Geosci., 19, 445–478, https://doi.org/10.1007/s10596-014-9443-x, 2015.

Stevens, N., Lehmann, C. E. R., Murphy, B. P., and Durigan, G.: Savanna woody encroachment is widespread across three continents, Glob. Change Biol., 23, 235–244, https://doi.org/10.1111/gcb.13409, 2017.

Steward, D. R., Yang, X., Lauwo, S. Y., Staggenborg, S. A., Macpherson, G. L., and Welch, S. M.: From precipitation to groundwater baseflow in a native prairie ecosystem: a regional study of the Konza LTER in the Flint Hills of Kansas, USA, Hydrol. Earth Syst. Sci., 15, 3181–3194, https://doi.org/10.5194/hess-15-3181-2011, 2011.

Sullivan, P. L., Stops, M. W., Macpherson, G. L., Li, L., Hirmas, D. R., and Dodds, W. K.: How landscape heterogeneity governs stream water concentration-discharge behavior in carbonate terrains (Konza Prairie, USA), Chem. Geol., 527, 118989, https://doi.org/10.1016/j.chemgeo.2018.12.002, 2019.

Tsypin, M. and Macpherson, G. L.: The effect of precipitation events on inorganic carbon in soil and shallow groundwater, Konza Prairie LTER Site, NE Kansas, USA, Appl. Geochem., 27, 2356–2369, https://doi.org/10.1016/j.apgeochem.2012.07.008, 2012.

Vargas, R., Collins, S. L., Thomey, M. L., Johnson, J. E., Brown, R. F., Natvig, D. O., and Friggens, M. T.: Precipitation variability and fire influence the temporal dynamics of soil $CO_2$ efflux in an arid grassland, Glob. Change Biol., 18, 1401–1411, https://doi.org/10.1111/j.1365-2486.2011.02628.x, 2012.

Vergani, C. and Graf, F.: Soil permeability, aggregate stability and root growth: a pot experiment from a soil bioengineering perspective, Ecohydrology, 9, 830–842, https://doi.org/10.1002/eco.1686, 2016.

Wang, J. A., Sulla-Menashe, D., Woodcock, C. E., Sonnentag, O., Keeling, R. F., and Friedl, M. A.: Extensive land cover change across Arctic-Boreal Northwestern North America from disturbance and climate forcing, Glob. Change Biol., 26, 807–822, https://doi.org/10.1111/gcb.14804, 2020.

Ward, D., Wiegand, K., and Getzin, S.: Walter's two-layer hypothesis revisited: back to the roots!, Oecologia, 172, 617–630, https://doi.org/10.1007/s00442-012-2538-y, 2013.

Watson, K. W. and Luxmoore, R. J.: Estimating macroporosity in a forest watershed by use of a tension infiltrometer, Soil Sci. Soc. Am. J., 50, 578–582, https://doi.org/10.2136/sssaj1986.03615995005000030007x, 1986.

Weiler, M. and Naef, F.: An experimental tracer study of the role of macropores in infiltration in grassland soils, Hydrol. Process., 17, 477–493, https://doi.org/10.1002/hyp.1136, 2003.

Wen, H. and Li, L.: An upscaled rate law for magnesite dissolution in heterogeneous porous media, Geochim. Cosmochim. Ac., 210, 289–305, https://doi.org/10.1016/j.gca.2017.04.019, 2017.

Wen, H. and Li, L.: An upscaled rate law for mineral dissolution in heterogeneous media: The role of time and length scales, Geochim. Cosmochim. Ac., 235, 1–20, https://doi.org/10.1016/j.gca.2018.04.024, 2018.

Wen, H., Li, L., Crandall, D., and Hakala, A.: Where lower calcite abundance creates more alteration: enhanced rock matrix diffusivity induced by preferential dissolution, Energy Fuels, 30, 4197–4208, https://doi.org/10.1021/acs.energyfuels.5b02932, 2016.

Wen, H., Perdrial, J., Abbott, B. W., Bernal, S., Dupas, R., Godsey, S. E., Harpold, A., Rizzo, D., Underwood, K., Adler, T., Sterle, G., and Li, L.: Temperature controls production but hydrology regulates export of dissolved organic carbon at the catchment scale, Hydrol. Earth Syst. Sci., 24, 945–966, https://doi.org/10.5194/hess-24-945-2020, 2020.

White, A. F. and Brantley, S. L.: The effect of time on the weathering of silicate minerals: why do weathering rates differ in the laboratory and field?, Chem. Geol., 202, 479–506, https://doi.org/10.1016/j.chemgeo.2003.03.001, 2003.

Winnick, M. J. and Maher, K.: Relationships between $CO_2$, thermodynamic limits on silicate weathering, and the strength of the silicate weathering feedback, Earth Planet. Sc. Lett., 485, 111–120, https://doi.org/10.1016/j.epsl.2018.01.005, 2018.

Wolery, T. J., Jackson, K. J., Bourcier, W. L., Bruton, C. J., Viani, B. E., Knauss, K. G., and Delany, J. M.: Current Status of the Eq3/6 Software Package for Geochemical Modeling, Acs. Sym. Ser., 416, 104–116, 1990.

Wu, Z. T., Dijkstra, P., Koch, G. W., Penuelas, J., and Hungate, B. A.: Responses of terrestrial ecosystems to temperature and precipitation change: a meta-analysis of experimental manipulation, Glob. Change Biol., 17, 927–942, https://doi.org/10.1111/j.1365-2486.2010.02302.x, 2011.

Zhang, Y. H., Niu, J. Z., Yu, X. X., Zhu, W. L., and Du, X. Q.: Effects of fine root length density and root biomass on soil preferential flow in forest ecosystems, For. Syst., 24, https://doi.org/10.5424/fs/2015241-06048, 2015. TS15

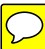

Zhi, W., Li, L., Dong, W. M., Brown, W., Kaye, J., Steefel, C. I., and Williams, K. H.: Distinct Source Water Chemistry Shapes Contrasting Concentration – Discharge Patterns, Water Resour. Res., 55, 4233–4251, https://doi.org/10.1029/2018wr024257, 2019.

Zhi, W. and Li, L.: The Shallow and Deep Hypothesis: Subsurface Vertical Chemical Contrasts Shape Nitrate Export Patterns from Different Land Uses, Environ. Sci. Technol., 54, 11915–11928, https://doi.org/10.1021/acs.est.0c01340, 2020.

Zhou, X. H., Talley, M., and Luo, Y. Q.: Biomass, Litter, and Soil Respiration Along a Precipitation Gradient in Southern Great Plains, USA, Ecosystems, 12, 1369–1380, https://doi.org/10.1007/s10021-009-9296-7, 2009.

Zhu, H., Zhang, L. M., and Garg, A.: Investigating plant transpiration-induced soil suction affected by root morphology and root depth, Comput. Geotech., 103, 26–31, https://doi.org/10.1016/j.compgeo.2018.06.019, 2018.

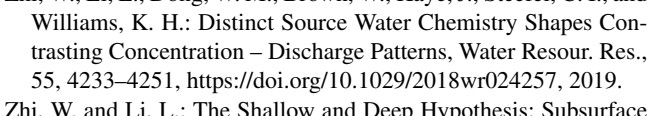

Please note the remarks at the end of the manuscript.

**Remarks from the language copy-editor**

CE1  Please note the slight edits to the format of the affiliations.

CE2  It is our standard to write out units in the text when not used in conjunction with a number (the instances set aside in parentheses are fine). I have adjusted this throughout. Please verify all instances.

CE3  Something appears to be missing here. Please provide a complete caption.

CE4  Please confirm added definition.

CE5  Do you mean complexes? Complexation is a verb (see https://www.merriam-webster.com/dictionary/complexation).

CE6  What does "them" refer to here?

**Remarks from the typesetter**

TS1  The composition of Figs. 1, 2, 4, 5 and 7 has been adjusted to our standards.

TS2  Please provide running title.

TS3  Please add last access date.

TS4  Please confirm link.

TS5  Please add last access date.

TS6  Aragão et al., 2009?   Yes

TS7  Please check panels in the caption.

TS8  Please check equations.

TS9  Please provide a reference list entry including creators, title, and date of last access.

TS10  Please confirm link.

TS11  Please provide a reference list entry including creators, title, and date of last access.

TS12  Please send a new supplement as a *.pdf without the title, authors, correspondence author, etc. as we will generate a supplement title page during publication (with a citation including the DOI), which will contain this information.

TS13  Please note that the funding information has been added to this paper. Please check if it is correct. Please also double-check your acknowledgements to see whether repeated information can be removed or changed accordingly. Thanks.  Checked

TS14  Please provide DOI.

TS15  Please provide article number or page range.