# Peer review of "Deepening roots can enhance carbonate weathering by amplifying CO2-enriched recharge"

_Biogeosciences, 2020_

## Referee Comment (RC1) · Anonymous Referee #1 · 17 Jun 2020

The manuscript ran detailed models to understand the effects of vegetation shift from grassland to woody plants on CaCO3 dissolution. The manuscript is well written and the conclusions are acceptable. Nevertheless, the model and calculations are far from my expertise and I am unable to judge them.

---

## Referee Comment (RC2) · Anonymous Referee #2 · 22 Jun 2020

The authors study the difference between chemical weathering in grassland and woodland soils through the application of simulations of a model calibrated in the grassland soil region. They aimed to understand the effect of deepening roots on carbonate weathering. In general, the manuscript is well written and structured; the geochemical model and simulation results are well described, using kinetic and thermodynamic equations. It deserves to be published because it deals with some relevant aspects that should be considered in studies dealing with the critical zone, which is the weathering of carbonate minerals and its link with the ecosystem; it also discusses the effects on the global carbon cycle and regional long-term simulations. Nevertheless, from what the title of the manuscript implies, I was expecting an extensive discussion about the deepening roots and their connection with weathering of carbonates, with

a well representation in the geochemical model. The authors did not create a model where it explicitly represent the rhizosphere nor biogeochemical processes besides respired CO2, despite that the last one was found to be one of the most important variables controlling weathering of carbonates. From my perspective, the deepening roots in the model were represented as pure hydrological feature, their only influence the flow path but not the flow itself, which can vary from plant species (not all plants will have the same water requirements). Although they discussed most of the before mentioned issues under model limitations, this subsection was somehow unclear. I would suggests a revision of the objectives or the model limitations in order to publish the manuscript. Specific comments: Why did the authors not include organic acids in the model or in the discussion? Is it because a small effect on weathering is expected? The long-term simulations displays a change in porosity, is this change considered in the short-term model? The calibration was done using grassland data, but the water uptake of plants depends also on the species, might it be the case that the roots take more water in woodland soils? Under model limitations, the flow partitioned is well explained, however, the points given above are not present, including the difference in water balance (water uptake by plants) among species. Technical comments: Units are usually separated from the number, including in percentage or permil; line 56-57, I did not understand completely this sentence; figure S1 should include coordinates in x and y axis; line 198, the symbol '>' may be changed into words; the unit for year is commonly written as 'a' not 'yr'.

---

## Referee Comment (RC3) · Anonymous Referee #3 · 23 Jun 2020

This manuscript deals with a topic of paramount importance in the present context of soil C research: the potential consequences of land-use changes (in this case, wood encroachment in grassland areas) in the soil inorganic C dynamics. It offers a comprehensive perspective and succeeds in providing some simulated data not only on the potential consequences of such change on soil $CO_2$ and water flows, but also on their relative importance as drivers of carbonates weathering in the study area. In this sense, it offers, within the limitations of the modelling approach considered and the available data, a quantitative view of the potential consequences of the mid-term changes in soil $pCO_2$ and water infiltration when vegetation changes in a soil with calcite accumulation at depth.

In my view, the manuscript is well written, describes well a complicated numerical

simulation (the materials in SI are very helpful), and contains interesting information, and a territory-based approach that makes it worth of interest.

The contribution is therefore meaningful and adds to a field where more information is needed. The quality of Figures is very high, both in terms of graphical design and the information they provide.

There are however some points that were not completely clear to me while reading the manuscript, and therefore I consider that they could be complemented for a clearer understanding. I cannot add much to the thermodynamical considerations in the construction of the model, but, as a soil scientist, there are some questions that came to my mind while revising the manuscript.

- In relation to the soil profile description and the gradient in calcite concentration depicted in Figure 1 ("The gray color gradient reflects the calcite abundance with more calcite in depth") and Figure 3.b, it is not clear for the reader which was the actual calcite distribution with depth. Only in the supplementary information (Table S1) it is said that the calcite volumetric concentration (m3/m3) was 0 in the upper soil horizon (0-54 cm). In line 324, the calcite-no calcite interface is set at 55 cm. If this is correct, this makes that in the model representing the grassland situation, where 95% of the flow was considered to "exit the soil column to the stream" at 50 cm (line 166-167), there would be almost no contact between soil water and calcite accumulated in the soil profile below 54 cm. Indeed (line 359), in the HF simulations, it is said that water "bypassed the calcite zone". I understand this can represent the actual situation in the study site, but, if this was the case, I believe this should be more clearly stated in the description of the site and the model: deeper rooting with wood encroachment would actually take soil water enriched in CO2 to the calcite accumulation areas in the soil profile, where otherwise (grassland), only a very limited amount of it would reach. I also wonder how realistic this can be if the actual distribution of rainfall along the year would be considered (for instance, intense storm episodes or wet vs. dry season rainfall patterns).

- In this sense, there is some confusion about the type of materials water enters in contact with at depth. While the abstract talks about "carbonate rocks" (line. 27), suggesting the parent material, the description of the soil profile (Table S1) denotes the horizon at 146-188 cm as "B" and that at 99-145 cm as "AB". This would mean that these would be accumulation horizons of materials leached from the upper "A" horizon (0-54 cm). If I understood well this table, their mineral composition according to this table indicates that calcite was not the dominant mineral in any of them.

- In relation to this, a point that is also not clear to me in the manuscript, is to what point re-precipitation of dissolved calcite at depth could determine some of the consequences of increased calcite dissolution with deeper woodland root systems. This would affect both the final fate of Ca and DIC in groundwater, and the assumptions done on the changes in permeability due to increased porosity with calcite depletion (lines 385-394). It is known that in this type of soils, carbonates can re-precipitate for instance around root channels, where water concentration can decrease rapidly compared to the bulk soil matrix.

- In relation to the annual average $CO_2(g)$ and $CO_2$ (aq) concentrations used in the numerical simulations (Table 1), and the idea explained in l.310-315 that some "impositions" on time and depth distribution of soil $CO_2$ had to be done to the model to capture the variation in alkalinity and Ca data and different horizons, I wonder to what extent this can be considered in the discussion about the use of annual averages to explain processes that can be very dependent of monthly (or even daily) fluctuations (in addition to spatial heterogeneity).

- Finally, a small comment on the term "shallow soil". It is used repeatedly in the manuscript to refer in fact to the model where grassland roots and subsurface waterflow are considered (in contrast to woodland roots and waterflow). For example, in l. 319 ("soil $CO_2$ production rate was the highest in the shallow soil"). In my understanding, the soil considered in the model has the same depth in all cases (Figure 1). This makes the term "shallow" (vs. deep?) misleading. I'd suggest to revise this point and

either change the term "shallow" and/or to use it to qualify the upper soil layer or the waterflow, not the soil.

---

## Referee Comment (RC4) · Anonymous Referee #4 · 30 Jun 2020

Compared to soil organic carbon, inorganic carbon in soil (SIC) is usually ignored in global carbon cycle. Increasing evidences indicates that SIC also plays important role in global carbon cycle. Therefore, this manuscript deals with a very interesting topic, i.e., effect of plants on carbonate weathering. However, I'm not an expert on model, and could not review it. After reading it, I feel that this manuscript has presented a very clear concept on the factors that control carbonate weathering, including temperature, hydrological regimes, and soil CO2 concentration. I just have a comment on it. The deep root systems of plants in some regions (e.g., semi-arid, and semi-humid) may make the deep soil dry, due to the strong transpiration.

---

## Referee Comment (RC5) · Anonymous Referee #5 · 6 Jul 2020

The manuscript titled "Deepening roots can enhance carbonate weathering" by Wen et al. documents the role that landscape changes (specifically changes from grasslands to woodlands) play in the dissolution of carbonate host rock. The manuscript is well written and addresses the important topic of flow partitioning- vs. soil CO2-driven weathering in carbonate terrain. The authors ran detailed model simulations to address their research questions:

1. "How and to what degree do rooting characteristics influence carbonate weathering when considering both flow partitioning and soil CO2 distribution?" 2. "Which factor (flow partitioning or soil CO2 distribution) predominantly controls weathering?"

From the title and abstract, I was expecting more discussion of rooting depth and its connection to weathering (as another reviewer points out). However, the paper focuses

on roots only as they relate to hydrologic flowpaths. Thus, much of the content and modeling is outside my area of expertise so I do not feel I can adequately judge the conclusions of the paper.

---

## Author Comment (AC1) · 12 Aug 2020

**Response to Reviewers' Comments**

We appreciate the efforts of the reviewers for their insightful and constructive comments. We have addressed concerns in the previous round of review. Below, we provide detailed, point-by-point responses to each of the reviewers' comments. We put the reviewer comments in regular font, author responses in blue, and direct quotes from the revised manuscript *in italic*.

**Reviewer #1' comments:**

The manuscript ran detailed models to understand the effects of vegetation shift from grassland to woody plants on CaCO3 dissolution. The manuscript is well written and the conclusions are acceptable. Nevertheless, the model and calculations are far from my expertise and I am unable to judge them.

Response:
> We appreciate the reviewer's comments.

**Reviewer #2' comments:**

The authors study the difference between chemical weathering in grassland and woodland soils through the application of simulations of a model calibrated in the grassland soil region. They aimed to understand the effect of deepening roots on carbonate weathering. In general, the manuscript is well written and structured; the geochemical model and simulation results are well described, using kinetic and thermodynamic equations. It deserves to be published because it deals with some relevant aspects that should be considered in studies dealing with the critical zone, which is the weathering of carbonate minerals and its link with the ecosystem; it also discusses the effects on the global carbon cycle and regional long-term simulations.

Nevertheless, from what the title of the manuscript implies, I was expecting an extensive discussion about the deepening roots and their connection with weathering of carbonates, with a well representation in the geochemical model. The authors did not create a model where it explicitly represent the rhizosphere nor biogeochemical processes besides respired CO2, despite that the last one was found to be one of the most important variables controlling weathering of carbonates. From my perspective, the deepening roots in the model were represented as pure hydrological feature, their only influence the flow path but not the flow itself, which can vary from plant species (not all plants will have the same water requirements). Although they discussed most of the before mentioned issues under model limitations, this subsection was somehow unclear. I would suggest a revision of the objectives or the model limitations in order to publish the manuscript.

Response:
> We appreciate the reviewer's encouraging comments. We agreed with the reviewer that the model does not explicitly include all processes related to deepening roots. To recognize this point and to avoid misleading the readers, we have changed the title to "Deepening roots can enhance carbonate weathering by amplifying recharge".

We also cleared the objectives and added more discussions about the model limitations, Line 112-114, 197-201, and 522-529:

*"Rooting characteristics can have multiple influences on water flow paths and water budget, for example, via water uptake and transpiration (Sadras, 2003;Fan et al., 2017;Pierret et al., 2016). This study focuses primarily on their potential influence via the alteration of hydrological flow paths."*

*"We compared the relative significance of hydrological and biogeochemical effects (soil $CO_2$ level and distribution) of rooting depths. Other conditions were assumed to be the same in the grassland and woodland numerical experiments so we can isolate the effects of hydrological flow path impacts."*

*"In addition to the alterations of hydrological flow paths, deepening roots may also affect the proportion of plant water uptake or soil water loss through transpiration (Pierret et al., 2016;Zhu et al., 2018), further modifying water fluxes into deep subsurface layers. Representing these dynamics requires a large number of processes that we do not have data to constrain. For example, studies of semiarid woodlands show that woody species predominantly use deep subsurface water while grasses predominantly use soil water from the upper soil layer (Ward et al., 2013). Furthermore, the grassland, with its shallower root distributions, may be more efficient in using water from small rainfall events than forests with their deeper root distributions (Mazzacavallo and Kulmatiski, 2015).Other studies have also documented significant competition for water uptake in upper soil layers among both woody and grass species (Scholes and Archer, 1997). It remains inconclusive how these water uptake characteristics are best represented in numerical experiments. These processes are therefore not included at this point ."*

Specific comments:

Why did the authors not include organic acids in the model or in the discussion? Is it because a small effect on weathering is expected?

Response:
We added the discussion about organic acids, Line 530-540:

*"In addition, distributions of microbes and organic acids associated with rooting structures vary with sites and seasons, and root channels and dry conditions with less soil water may trigger calcite re-precipitation. Microbial activities surrounding living and dead roots can also lead to calcite precipitation and infilling of fractures and other macropores, and alter flow pathways (Lambers et al., 2009). Organic acids can decrease soil water pH, increase mineral solubilities through organic-metal complexations, and accelerate chemical weathering (Pittman and Lewan, 2012;Lawrence et al., 2014). Their impacts on carbonate weathering kinetics might be smaller (due to the fast kinetics) compared to their alteration of calcite solubility in natural systems. Such processes further complicate how to represent biogeochemical processes in models. Although we do not explicitly simulate all of these simultaneous and competing processes in our numerical experiments, the model was constrained by the field data in grasslands and woodland that have already integrated these effects in natural systems. Indeed, the weathering rates can be understood as the net weathering rates that are the net difference between dissolution and*

*reprecipitation. This is reflected in lower carbonate weathering rates under low infiltration (low flow) conditions."*

The long-term simulations displays a change in porosity, is this change considered in the short-term model?

Response:

Yes, this is also considered in the short-term model. However, the dissolved calcite volume in the short-term numerical experiments (Scenarios 1-3) is less than 0.5% v/v and has minimal impact on permeability and water-rock contact time in these short-term numerical experiments. We clarified the changes of the solid phase in Line 293-294 and 307-308:

*"Note that numerical experiments in Scenario 1-3 focused on the short-term scale with negligible changes in the solid phase."*

*"The dissolved mineral volume was negligible (< 0.5% v/v)."*

The calibration was done using grassland data, but the water uptake of plants depends also on the species, might it be the case that the roots take more water in woodland soils?

Response:

We agreed that the water uptake may change with plant species and added discussions about the potential impact, Line 522-529:

*"In addition to the alterations of hydrological flow paths, deepening roots may also affect the proportion of plant water uptake or soil water loss through transpiration (Pierret et al., 2016;Zhu et al., 2018), further modifying water fluxes into deep subsurface layers. Representing these dynamics require a large number of processes that we do not have data to constrain. For example, studies of semiarid woodlands show that woody species predominantly use deep subsurface water while grasses predominantly use soil water from the upper soil layer (Ward et al., 2013). Furthermore, the grassland, with its shallower root distributions, may be more efficient in using water from small rainfall events than forests with their deeper root distributions (Mazzacavallo and Kulmatiski, 2015).Other studies have also documented significant competition for water uptake in upper soil layers among both woody and grass species (Scholes and Archer, 1997). It remains inconclusive how these water uptake characteristics are best represented in numerical experiments. These processes are therefore not included at this point ."*

Under model limitations, the flow partitioned is well explained, however, the points given above are not present, including the difference in water balance (water uptake by plants) among species.

Response:

We added more discussions about the organic acid and other potential differences among species in the section of model limitations, Line 522-529 and 530-540:

*"In addition to the alterations of hydrological flow paths, deepening roots may also affect the proportion of plant water uptake or soil water loss through transpiration (Pierret et al., 2016;Zhu et al., 2018), further modifying water fluxes into deep subsurface layers. Representing these dynamics requires a large number of processes that we do not have data to constrain. For example, studies of semiarid woodlands show that woody species predominantly use deep subsurface water while grasses predominantly use soil water from the upper soil layer (Ward et al., 2013). Furthermore, the grassland, with its shallower root distributions, may be more efficient in using water from small rainfall events than forests with their deeper root distributions (Mazzacavallo and Kulmatiski, 2015).Other studies have also documented significant competition for water uptake in upper soil layers among both woody and grass species (Scholes and Archer, 1997). It remains inconclusive how these water uptake characteristics are best represented in numerical experiments. These processes are therefore not included at this point ."*

*"In addition, distributions of microbes and organic acids associated with rooting structures vary with sites and seasons, and root channels and dry conditions with less soil water may trigger calcite re-precipitation. Microbial activities surrounding living and dead roots can also lead to calcite precipitation and infilling of fractures and other macropores, and alter flow pathways (Lambers et al., 2009). Organic acids can decrease soil water pH, increase mineral solubilities through organic-metal complexations, and accelerate chemical weathering (Pittman and Lewan, 2012;Lawrence et al., 2014). Their impacts on carbonate weathering kinetics might be smaller (due to the fast kinetics) compared to their alteration of calcite solubility in natural systems. Such processes further complicate how to represent biogeochemical processes in models. Although we do not explicitly simulate all of these simultaneous and competing processes in our numerical experiments, the model was constrained by the field data in grasslands and woodland that have already integrated these effects in natural systems. Indeed, the weathering rates can be understood as the net weathering rates that are the net difference between dissolution and reprecipitation. This is reflected in lower carbonate weathering rates under low infiltration (low flow) conditions."*

Technical comments:

Units are usually separated from the number, including in percentage or permil;

Response:
We modified the units through the whole manuscript.

line 56-57, I did not understand completely this sentence;

Response:
We modified the sentence, Line 57-58.

*"In grasslands, the lateral, dense spread of roots in upper soil layers promotes the formation of horizontally-oriented macropores that support near-surface lateral flow (Cheng et al., 2011)."*

figure S1 should include coordinates in x and y axis;

Response:

We included coordinates in Figure S1.

line 198, the symbol '>' may be changed into words;

Response:
We modified it into words, Line 206-207:

*"Deeper roots in woodlands can increase deep soil permeability by over one order of magnitude (Vergani and Graf, 2016)."*

the unit for year is commonly written as 'a' not 'yr'.

Response:
We revised the unit of year into 'a' through the whole manuscript.

**Reviewer #3' comments:**

This manuscript deals with a topic of paramount importance in the present context of soil C research: the potential consequences of land-use changes (in this case, wood encroachment in grassland areas) in the soil inorganic C dynamics. It offers a comprehensive perspective and succeeds in providing some simulated data not only on the potential consequences of such change on soil CO2 and water flows, but also on their relative importance as drivers of carbonates weathering in the study area. In this sense, it offers, within the limitations of the modelling approach considered and the available data, a quantitative view of the potential consequences of the mid-term changes in soil pCO2 and water infiltration when vegetation changes in a soil with calcite accumulation at depth.

In my view, the manuscript is well written, describes well a complicated numerical simulation (the materials in SI are very helpful), and contains interesting information, and a territory-based approach that makes it worth of interest.

The contribution is therefore meaningful and adds to a field where more information is needed. The quality of Figures is very high, both in terms of graphical design and the information they provide.

Response:
We appreciate the encouraging comments and the constructive suggestions below from the reviewer.

There are however some points that were not completely clear to me while reading the manuscript, and therefore I consider that they could be complemented for a clearer understanding. I cannot add much to the thermodynamical considerations in the construction of the model, but, as a soil scientist, there are some questions that came to my mind while revising the manuscript.

- In relation to the soil profile description and the gradient in calcite concentration depicted in Figure 1 ("The gray color gradient reflects the calcite abundance with more calcite in depth") and Figure 3.b, it is not clear for the reader which was the actual calcite distribution with depth. Only in the supplementary information (Table S1) it is said that the calcite volumetric concentration (m3/m3) was 0 in the upper soil horizon (0-54 cm). In line 324, the calcite-no calcite interface is set at 55 cm. If this is correct, this makes that in the model representing the grassland situation, where 95% of the flow was considered to "exit the soil column to the stream" at 50 cm (line 166-167), there would be almost no contact between soil water and calcite accumulated in the soil profile below 54 cm. Indeed (line 359), in the HF simulations, it is said that water "bypassed the calcite zone". I understand this can represent the actual situation in the study site, but, if this was the case, I believe this should be more clearly stated in the description of the site and the model: deeper rooting with wood encroachment would actually take soil water enriched in CO2 to the calcite accumulation areas in the soil profile, where otherwise (grassland), only a very limited amount of it would reach. I also wonder how realistic this can be if the actual distribution of rainfall along the year would be considered (for instance, intense storm episodes or wet vs. dry season rainfall patterns).

Response:

We have Figure 3A showing the depth profile of calcite volume used in all numerical experiments, consistent with the values listed in Table S1. We added clarifications in Line 229:

*"b. Calcite distribution (black line) in all cases increases with depth; The values are listed in Table S1 and shown in Figure 3A."*

We also added clarifications about how water flows through the deep calcite-abundant subsurface in the grassland and woodland cases, Line 70-71 and 105-109:

*"We hypothesized that deepening roots in woodlands enhance carbonate weathering by promoting deeper recharge and $CO_2$-carbonate contact in the deep, carbonate-abundant subsurface (Figure 1)."*

*"The shallow and dense fine roots in the grassland promote lateral macropore development and lateral water flow. In contrast, the woodlands induce vertical macropore development that supports vertical flow (recharge) into the deep, calcite-abundant subsurface compared to the grassland."*

This work aimed to compare the general, averaged behaviors rather than event-scale dynamics. We agreed that the actual distribution of rainfall may affect how and to what extent water reacts with calcite, and added discussions, Line 515-521:

*"There are many examples of mechanisms within soil profiles that can influence flow partitioning and are not explicitly represented in our modeling exercises. For example, contrasts in flow-conducting properties (i.e., porosity and permeability) in shallow and deep zones, physical and chemical heterogeneity in carbonate distribution (Wen and Li, 2018), connectivity between different areas of the catchment in dry and wet times (Wen et al., 2020), and water table and corresponding lateral flow depth associated with the rainfall frequency and intensity (Li et al.,*

*2017;Harman and Cosans, 2019). All these factors may affect the water-calcite contact extent, leading to the changes of carbonate weathering rates (Figure 6)."*

- In this sense, there is some confusion about the type of materials water enters in contact with at depth. While the abstract talks about "carbonate rocks" (line. 27), suggesting the parent material, the description of the soil profile (Table S1) denotes the horizon at 146-188 cm as "B" and that at 99-145 cm as "AB". This would mean that these would be accumulation horizons of materials leached from the upper "A" horizon (0-54 cm). If I understood well this table, their mineral composition according to this table indicates that calcite was not the dominant mineral in any of them.

Response:

We clarified it and added more descriptions about the mineralogy in the methodology, Line 27-28, 85-89 and 160-162:

*"These differences in weathering fronts are ultimately caused by the contact time of $CO_2$-charged water with abundant carbonate minerals in the deep subsurface."*

*"The bedrock contains repeating Permian couplets of limestone (1-2 m thick) and mudstone (2-4 m) (Macpherson et al., 2008). The limestone is primarily calcite with traces of dolomite while the mudstone is dominated by illite, chlorite, and mixed-layers of chlorite-illite and chlorite-vermiculite, varying in abundance from major to trace amounts. With an average thickness of 1-2 m in the lowlands, the soil mostly has a carbonate content of less than 25% (Macpherson et al., 2008)."*

*"In the model, the upper soil layers have more anorthite ($CaAl_2Si_2O_8(s)$) and K-feldspar ($KAlSi_3O_8(s)$) and the deeper subsurface contains more calcite (Table S1). The calcite volume increases from 0% in the upper soil layer to 10% in the deep subsurface."*

- In relation to this, a point that is also not clear to me in the manuscript, is to what point re-precipitation of dissolved calcite at depth could determine some of the consequences of increased calcite dissolution with deeper woodland root systems. This would affect both the final fate of Ca and DIC in groundwater, and the assumptions done on the changes in permeability due to increased porosity with calcite depletion (lines 385-394). It is known that in this type of soils, carbonates can re-precipitate for instance around root channels, where water concentration can decrease rapidly compared to the bulk soil matrix.

Response:

We agree that calcite reprecipitation can happen. And they often happen under dry conditions where soils are dried out. This work focuses on average behavior and the first-order principles of hydrological ramification of roots. The model does not explicitly simulate details of root channels and dry conditions that may trigger calcite re-precipitation. However, the model was constrained by the field data in grasslands and woodland that have already integrated these effects in natural systems. In fact, the weathering rates can be understood as the net weathering rates that are the net difference between dissolution and reprecipitation. That is why carbonate weathering

rates are lower under low infiltration (low flow) conditions. We discussed the potential impacts of calcite re-precipitation in the section of model limitations, Line 530-540:

*"In addition, distributions of microbes and organic acids associated with rooting structures vary with sites and seasons, and root channels and dry conditions with less soil water may trigger calcite re-precipitation. Microbial activities surrounding living and dead roots can also lead to calcite precipitation and infilling of fractures and other macropores, and alter flow pathways (Lambers et al., 2009). Organic acids can decrease soil water pH, increase mineral solubilities through organic-metal complexations, and accelerate chemical weathering (Pittman and Lewan, 2012;Lawrence et al., 2014). Their impacts on carbonate weathering kinetics might be smaller (due to the fast kinetics) compared to their alteration of calcite solubility in natural systems. Such processes further complicate how to represent biogeochemical processes in models. Although we do not explicitly simulate all of these simultaneous and competing processes in our numerical experiments, the model was constrained by the field data in grasslands and woodland that have already integrated these effects in natural systems. Indeed, the weathering rates can be understood as the net weathering rates that are the net difference between dissolution and reprecipitation. This is reflected in lower carbonate weathering rates under low infiltration (low flow) conditions."*

- In relation to the annual average CO2(g) and CO2 (aq) concentrations used in the numerical simulations (Table 1), and the idea explained in l.310-315 that some "impositions" on time and depth distribution of soil CO2 had to be done to the model to capture the variation in alkalinity and Ca data and different horizons, I wonder to what extent this can be considered in the discussion about the use of annual averages to explain processes that can be very dependent of monthly (or even daily) fluctuations (in addition to spatial heterogeneity).

Response:

The numerical experiments in this work focus on the general behaviors using the annual average $CO_2(g)$, which is the dominant driver of Ca and DIC concentrations in soil water. At daily or monthly scales, the variations of rainfall frequency and intensity may change the flow conditions (e.g., water table and lateral flow depth), leading to different extent of water-calcite contact extent and weathering rates. We added discussions about it, Line 515-521:

*"There are many examples of mechanisms within soil profiles that can influence flow partitioning and are not explicitly represented in our modeling exercises. For example, contrasts in flow-conducting properties (i.e., porosity and permeability) in shallow and deep zones, physical and chemical heterogeneity in carbonate distribution (Wen and Li, 2018), connectivity between different areas of the catchment in dry and wet times (Wen et al., 2020), and water table and corresponding lateral flow depth associated with the rainfall frequency and intensity (Li et al., 2017;Harman and Cosans, 2019). All these factors may affect the water-calcite contact extent, leading to the changes of carbonate weathering rates (Figure 6)."*

- Finally, a small comment on the term "shallow soil". It is used repeatedly in the manuscript to refer in fact to the model where grassland roots and subsurface waterflow are considered (in contrast to woodland roots and waterflow). For example, in l. 319 ("soil CO2 production rate was the highest in the shallow soil"). In my understanding, the soil considered in the model has the

same depth in all cases (Figure 1). This makes the term "shallow" (vs. deep?) misleading. I'd suggest to revise this point and either change the term "shallow" and/or to use it to qualify the upper soil layer or the waterflow, not the soil.

Response:
We appreciate this comment. We have modified the terms into "upper soil layer" to avoid confusion.

**Reviewer #4' comments:**

Compared to soil organic carbon, inorganic carbon in soil (SIC) is usually ignored in global carbon cycle. Increasing evidences indicates that SIC also plays important role in global carbon cycle. Therefore, this manuscript deals with a very interesting topic, i.e., effect of plants on carbonate weathering. However, I'm not an expert on model, and could not review it. After reading it, I feel that this manuscript has presented a very clear concept on the factors that control carbonate weathering, including temperature, hydrological regimes, and soil CO2 concentration. I just have a comment on it. The deep root systems of plants in some regions (e.g., semi-arid, and semi-humid) may make the deep soil dry, due to the strong transpiration.

Response:
We thank the reviewer for reading the manuscript. We added discussions about water uptake, Line 522-529:

*"In addition to the alterations of hydrological flow paths, deepening roots may also affect the proportion of plant water uptake or soil water loss through transpiration (Pierret et al., 2016;Zhu et al., 2018), further modifying water fluxes into deep subsurface layers. Representing these dynamics requires a large number of processes that we do not have data to constrain. For example, studies of semiarid woodlands show that woody species predominantly use deep subsurface water while grasses predominantly use soil water from the upper soil layer (Ward et al., 2013). Furthermore, the grassland, with its shallower root distributions, may be more efficient in using water from small rainfall events than forests with their deeper root distributions (Mazzacavallo and Kulmatiski, 2015).Other studies have also documented significant competition for water uptake in upper soil layers among both woody and grass species (Scholes and Archer, 1997). It remains inconclusive how these water uptake characteristics are best represented in numerical experiments. These processes are therefore not included at this point ."*

**Reviewer #5' comments:**

The manuscript titled "Deepening roots can enhance carbonate weathering" by Wen et al. documents the role that landscape changes (specifically changes from grasslands to woodlands) play in the dissolution of carbonate host rock. The manuscript is well written and addresses the important topic of flow partitioning- vs. soil CO2-driven weathering in carbonate terrain. The authors ran detailed model simulations to address their research questions:

1. "How and to what degree do rooting characteristics influence carbonate weathering when considering both flow partitioning and soil CO2 distribution?"
2. "Which factor (flow partitioning or soil CO2 distribution) predominantly controls weathering?"

From the title and abstract, I was expecting more discussion of rooting depth and its connection to weathering (as another reviewer points out). However, the paper focuses on roots only as they relate to hydrologic flowpaths. Thus, much of the content and modeling is outside my area of expertise so I do not feel I can adequately judge the conclusions of the paper.

Response:

We thank the reviewer for reading the manuscript. Based on comments from all of the reviewers, including Reviewer #5, we have chosen to change the title of the manuscript:

*"Deepening roots can enhance carbonate weathering by amplifying recharge"*

---

## Author Comment (AC2) · 12 Aug 2020

Please see attachment.

Please also note the supplement to this comment:
https://bg.copernicus.org/preprints/bg-2020-180/bg-2020-180-AC2-supplement.pdf

---

## Author Comment (AC3) · 12 Aug 2020

Please see attachment.

Please also note the supplement to this comment:
https://bg.copernicus.org/preprints/bg-2020-180/bg-2020-180-AC3-supplement.pdf

---

## Author Comment (AC5) · 12 Aug 2020

Please see attachment.

Please also note the supplement to this comment:
https://bg.copernicus.org/preprints/bg-2020-180/bg-2020-180-AC5-supplement.pdf